# The small iron-deficiency-induced protein OLIVIA and its relation to the bHLH transcription factor POPEYE

**Daniela M. Lichtblau**[1☉], **Dibin Baby**[1☉], **Mather Khan**[1,2☉], **Ksenia Trofimov**[1], **Yunus Ari**[1], **Birte Schwarz**[1], **Petra Bauer**[1,2]*

**1** Institute of Botany, Heinrich Heine University, Düsseldorf, Germany, **2** Cluster of Excellence on Plant Science (CEPLAS), Heinrich Heine University, Düsseldorf, Germany

☉ These authors contributed equally to this work.
* petra.bauer@hhu.de

**Data Availability Statement:** All relevant data are within the manuscript and its Supporting Information files.

## Abstract

Iron (Fe) is a crucial micronutrient needed in many metabolic processes. To balance needs and potential toxicity, plants control the amount of Fe they take up and allocate to leaves and seeds during their development. One important regulator of this process is POPEYE (PYE). PYE is a Fe deficiency-induced key bHLH transcription factor (TF) for allocation of internal Fe in plants. In the absence of PYE, there is altered Fe translocation and plants develop a leaf chlorosis. *NICOTIANAMINE SYNTHASE4* (*NAS4*), *FERRIC-REDUCTION OXIDASE3* (*FRO3*), and *ZINC-INDUCED FACILITATOR1* (*ZIF1*) genes are expressed at higher level in *pye-1* indicating that PYE represses these genes. PYE activity is controlled in a yet unknown manner. Here, we show that a small Fe deficiency-induced protein OLIVIA (OLV) can interact with PYE. OLV has a conserved C-terminal motif, that we named TGIYY. Through deletion mapping, we pinpointed that OLV TGIYY and several regions of PYE can be involved in the protein interaction. An *OLV* overexpressing (OX) mutant line exhibited an enhanced *NAS4* gene expression. This was a mild Fe deficiency response phenotype that was related to PYE function. Leaf rosettes of *olv* mutants remained smaller than those of wild type, indicating that OLV promotes plant growth. Taken together, our study identified a small protein OLV as a candidate that may connect aspects of Fe homeostasis with regulation of leaf growth.

## Introduction

Iron (Fe) is a cofactor in many redox reactions and serves multiple functions in fundamental metabolic processes such as photosynthesis and cell respiration. Despite of being essential, Fe can become toxic and promote oxidative stress when it is present in excess. Thus, the micronutrient Fe is a decisive element for plant growth and health, and plants control tightly Fe acquisition and allocation.

Despite the fact that Fe is the fourth most abundant element in the continental crust, most of it is present in the form of poorly soluble $Fe^{3+}$ oxides in the soil [1]. Inside plants, Fe can be

**Funding:** This work was funded by Deutsche Forschungsgemeinschaft (DFG, German Research Foundation) under GRK F020512056 (NEXTplant) and Germany´s Excellence Strategy – EXC-2048/1 – project ID 390686111. Funding for instrumentation: Zeiss LSM780 + 4-channel FLIM extension (Picoquant): DFG- INST 208/551-1 FUGG. The funders had no role in study design, data collection and analysis, decision to publish, or preparation of the manuscript.

**Competing interests:** The authors have declared that no competing interests exist.

**Abbreviations:** bHLH, Basic helix-loop-helix; BiFC, Bimolecular fluorescence complementation; FRET, Förster Resonance Energy Transfer After Photobleaching; GFP, Green fluorescence protein; GUS, ß-Glucuronidase; mCherry, Second generation mRFP derivate; mRFP, Monomeric red fluorescence protein; OLV, OLIVIA; OX, Overexpression; PYE, POPEYE; RT-qPCR, Reverse transcription quantitative PCR; SD, Standard deviation; TF, Transcription factor; WT, Wild type; Y2H, Yeast two-hybrid; YFP, Yellow fluorescence protein.

bound or stored by cell walls, plastidial ferritin or precipitates in the vacuole. Plants have a number of transport proteins and enzymes that help them mobilize Fe and allocate Fe across membranes and long-distance, from soil to root epidermis cells, across different tissues, from roots to leaves and from there to sink organs. These transport processes may involve Fe reduction and chelation e.g. by ferric reductase oxidases and nicotianamine [2]. Fe homeostasis of plants depends on the available Fe sources in the soil, but also on the plants' requirements for Fe during growth. When plants sense Fe deficiency, they mobilize Fe in the soil and within the plants. This is controlled by a cascade of bHLH transcription factors (TFs) that are activated by low Fe. Different subgroups of bHLH TFs steer different aspects of Fe utilization in leaves and roots, and their activities are positively and negatively controlled [3]. Yet, how the sensing of Fe deficiency is coupled with plant growth and the required uptake and delivery of Fe to target sites bares still many open questions. There are open questions how the bHLH TFs are controlled and which role positive and negative regulation plays.

POPEYE (PYE) is a Fe deficiency-induced bHLH TF, that mediates the internal mobilization of Fe in roots and shoots. PYE is most strongly expressed in the root stele and vascular bundles in leaves although it seems prone to cell-to-cell movement. In the absence of PYE, plants fail to fully utilize Fe and develop a Fe deficiency leaf chlorosis [4]. *pye* loss of function mutants are not able to mobilize internal Fe. At the same time, they show up-regulated expression of different target genes [4,5]. PYE target genes include *FRO3*, *NAS4* and *ZIF1* [4]. FRO3 is a Fe-chelate reductase involved in mitochondrial Fe import [6]. Nicotianamine synthase (NAS4) can produce a metal chelator, nicotianamine, needed for metal ion allocation [7,8]. This nicotianamine may serve to translocate internal cellular metal ions towards the vacuole. Indeed, zinc facilitator ZIF1 is a vacuolar nicotianamine-metal importer [9]. Perhaps, *pye* mutants retain metal ions in the vacuole instead of mobilizing them to various sink organs upon Fe deficiency. This suggests that PYE may prevent the internal cellular Fe and metal ion storage machinery in the mitochondria and vacuole which then promotes allocation of Fe to other leaves. PYE responds to Fe deficiency signals. It can interact with PYE-LIKE homologs of the bHLH subgroup IVc group (e.g. bHLH104/105/115) [4,10]. It can also form homodimers [5,11]. A recent study showed that PYE may repress *PYE* itself and its co-expressed *BHLH* subgroup Ib genes [5]. Hence, this varying effect of PYE on different genes indicates that a molecular mechanism must be present that regulates PYE activity to maintain system homeostasis. PYE and related bHLH TFs can interact in protein complexes that may fine-tune the Fe deficiency response, and harbor for example Fe deficiency-induced E3 ligases BRUTUS (BTS)/ BTS-LIKE (BTSL) as negative factors [11,12].

Small proteins are defined as polypeptides with usually less than 100 amino acids (aa) [13]. They are often mobile and can act in nutrient-related signaling [14]. In addition, they can be involved in the regulation of protein activity [15]. A prominent class of protein interactors are small plant protein effectors that are injected by pathogens into plant cells to dampen the plant immune responses by interacting with proteins of many different kinds [16]. Small proteins can interact and prevent TF complexes from functioning, thus inhibiting their transcriptional regulation activities [17]. For example, KIDARI (KDR) is a small protein with Helix-Loop-Helix (HLH) domain that may act as a repressor of light signaling in Arabidopsis by binding to and inhibiting the activity of a light-controlled bHLH TF [18]. Another example, BBX31 (length of 121 amino acids) and BBX30 (length of 117 amino acids) allosterically deactivate TFs [19,20]. Therefore, identifying and characterizing small protein candidates that modulate bHLH TF activity in the Fe signaling pathway can be relevant for better understanding the molecular mechanisms involved in PYE regulation. Presently, the importance of small effector proteins interfering with Fe deficiency responses is far from being fully understood. It is interesting to note that there are several Fe-deficiency-induced small proteins. Best studied among

them is the family of IRON MAN (IMA) small proteins, also known as FE UPTAKE-INDU-CING PEPTIDEs (FEPs) [21,22]. IMA small proteins mechanistically regulate the iron homeo-stasis signaling pathway by attenuating interactions between BTS or BTSLs and bHLH IVc TFs through direct binding, thereby stabilizing bHLH IVc TFs. PYE differs structurally from related bHLH TFs with regard to the protein interaction sites for BTS/Ls [11]. PYE lacks the conserved BTS interaction motif. Therefore, evidence is still pending regarding the effect of BTS on modulating PYE at the protein level. Consequently, the existence of a different regula-tor of PYE function stands out as salient and warrants further investigation.

Here, we describe that PYE can bind to an Fe deficiency-inducible small protein that we named OLIVIA (OLV). We delimited the protein interaction sites. A molecular-physiological analysis suggests that OLV has only mild effects on PYE-dependent Fe deficiency responses. OLV also affects rosette size and plant growth, indicating a link between Fe deficiency responses during plant growth.

## Results

### PYE interacted with OLV in yeast and in plant cells

Protein-protein interactions play a crucial role to fine-tune the regulation of Fe uptake and homeostasis, and many of the relevant protein interactions for Fe-regulatory processes involve co-expressed proteins. We have previously reported that co-expression networks help identify-ing novel protein-protein interactions [11]. Among the tested co-expressed proteins was At1g73120 (**S1 Fig**). At1g73120 encodes a small protein which we named OLIVIA (OLV) according to its partnership with PYE. PYE and OLV were co-transformed into yeast and tested in targeted yeast two hybrid (Y2H) assays. These assays allowed quantitative conclusions about the interaction strength, when yeast serial dilutions were analyzed. The interactions between PYE and OLV were re-confirmed in reciprocal manner (**Fig 1A**).

Next, we verified this interaction in plant protein interaction assays by using Förster reso-nance energy transfer-acceptor photobleaching (FRET-APB) and bimolecular fluorescence complementation (BiFC). FRET-APB is a sensitive method to analyze protein interactions, and FRET efficiency can be used to obtain a quantitative measurement for the interaction strength [23]. FRET-APB analysis was performed in plant nuclei. FRET was detected for the pair PYE-GFP and OLV-mCherry with a FRET efficiency of 3.5%. This FRET efficiency was significantly higher than the negative control (PYE-GFP only) (**Fig 1B**), confirming an interac-tion of PYE and OLV. Controlled BiFC experiments [23,24] provided additional hints. A YFP signal was obtained when both nYFP-PYE and cYFP-OLV fusion proteins were expressed in cells. BiFC indicated that PYE and OLV interacted in the nucleus and cytoplasm (**Fig 1C**). Negative controls were nYFP-PYE and cYFP-OLV together with either cYFP-bHLH039 or nYFP-ILR3, that did not result in YFP signals (**Fig 1C**).

In summary, PYE and OLV can interact in yeast and plant cells, as verified by three inde-pendent methods.

### OLV carries a conserved TGIYY motif that facilitated interaction with PYE

To further prove the interaction capacities of OLV protein and describe it at a molecular level, we first examined OLV protein structural predictions and then conducted a mapping of inter-action sites for the OLV-PYE pair. *OLV* is a unique gene in the Arabidopsis genome. Ortho-logs of OLV are present in all investigated angiosperms. Interestingly, irrespective of the size and total percentage of amino acid (aa) identity, all OLV orthologs exhibited a conserved motif in the C-terminus (in AtOLV aa 71–87), which we named TGIYY motif due to the pres-ence of the respective five aa in the middle of the motif (WVPHEG**TGIYY**PKGQEK; **S2A,**

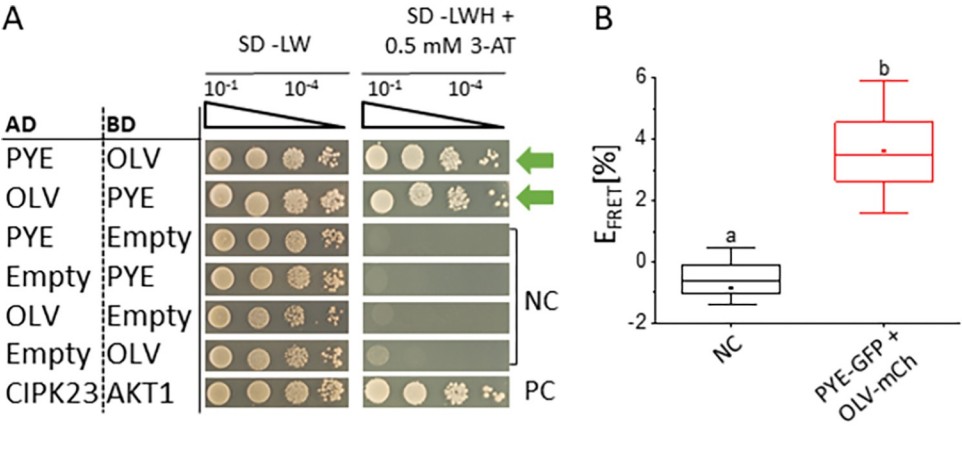

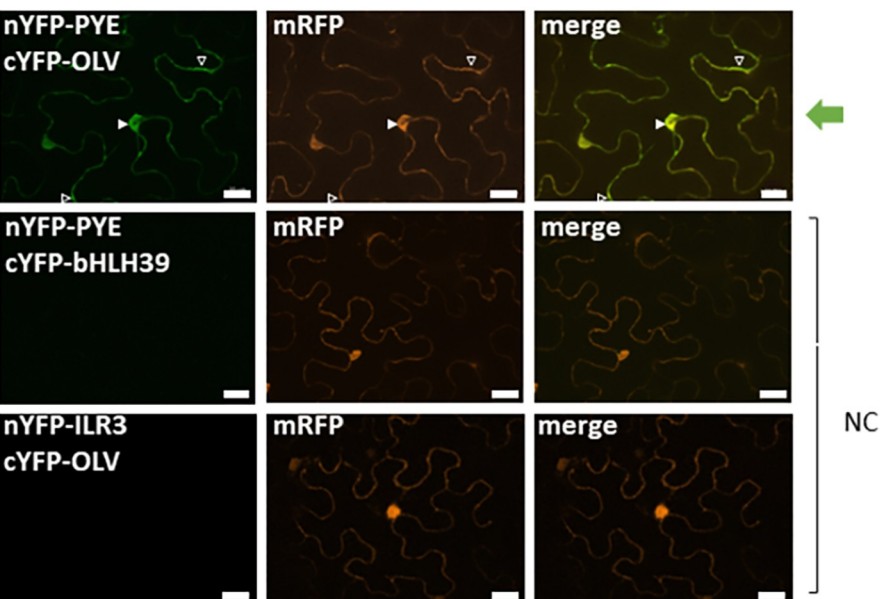

**Fig 1. PYE interacts with OLV in yeast and plant cells. (A)** Reciprocal targeted yeast two hybrid (Y2H) assay indicating OLV and PYE protein interaction. Yeast co-transformed with activation domain (AD) and binding domain (BD) plasmid combinations were spotted in 10-fold dilution series ($A_{600} = 10^{-1}$–$10^{-4}$) on SD-LW (positive growth and co-transformation control) and SD-LWH + 0.5 mM 3-AT (selection for protein interaction) plates. Negative controls (NC): Empty AD with BD protein; AD protein with empty BD. Positive control (PC): CIPK23 and cAKT1 (Xu et al., 2006). **(B)** Quantitative Förster resonance energy transfer-acceptor photobleaching (FRET-APB) measurements in transiently transformed tobacco leaf epidermis cells. Box plots indicate FRET efficiency ($E_{FRET}$) and strength of the interaction, determined in the nucleus using the pair PYE-GFP and OLV-mCherry (mCh). NC: Donor-only PYE-GFP. An increased FRET efficiency ($E_{FRET}$) of the FRET pair compared to the NC is an indication for protein interaction. Different letters indicate statistically significant differences (one-way ANOVA and Tukey´s post-hoc test, $p < 0.05$), n = 15 nuclei. **(C)** Bimolecular fluorescence complementation (BiFC) experiment in transiently transformed tobacco leaf epidermis cells, showing interaction between nYFP-PYE and cYFP-OLV. NC: nYFP-PYE + cYFP-bHLH39 and nYFP-ILR3 + cYFP-OLV. mRFP was used as transformation control. Complemented YFP signal indicates a protein interaction. Arrowheads indicate YFP signal. Scale bars: 20 μm. Green arrows indicate positive protein interaction.

S2B, S3A and S3B Figs). This motif does not resemble any known protein domain according to e.g Uniprot (www.uniprot.org) or InterPro-EMBL-EBI (www.ebi.ac.uk). Unlike Arabidopsis, some species have more than one ortholog of OLV or TGIYY-like-containing protein (for

example: *Vitis vinifera*, *Populus trichocarpa*, *Eucalyptus grandis*, *Sesamum indicum*, *Daucus carota*). TGIYY-like motif-containing proteins are also encoded in the genomes of green algae and lower land plants (**S3C Fig**), as well as in organisms of other kingdoms such as bacteria, fungi, animals (examples are represented in **S3D Fig**). In non-plant species, the TGIYY-like motif was part of proteins with unknown function that were much bigger than 109 aa (the sizes varied from less than hundred to several hundred aa).

In summary, all plants and other organisms have genes encoding proteins with a TGIYY or TGIYY-like motif. The high level of conservation points to a functional relevance of the TGIYY motif.

To investigate whether the conserved TGIYY motif is relevant for protein-protein interaction with PYE, we constructed OLV deletion mutant forms (**Fig 2A**) and tested them in BiFC assays. None of the OLV deletion variants devoid of TGIYY, namely OLV-N and OLV-DT, interacted with PYE in BiFC, while constructs harboring the TGIYY motif did, OLV-C and TGIYY (**Fig 2B**). OLV-N, OLV-C and OLV-DT were similarly localized in plant cell nuclei and the cytoplasm as OLV (**S4 Fig**). OLV-DT was also tested in Y2H and FRET-APB assays. While OLV-DT did not interact with PYE in Y2H, it was similarly capable of interacting with PYE in FRET-APB as the full-length OLV construct did (**Fig 2C and 2D**).

Hence, the TGIYY motif was important in two protein interaction assays but not in FRET-APB (**Fig 2E**). It may be that several regions of OLV can be involved in the interaction with PYE protein, which can be revealed depending on the tags of fusion proteins. We thus conclude that even through the TGIYY may not be essential it may facilitate the interaction between OLV and PYE.

## Different regions of PYE were involved in protein interaction with OLV

Next, we determined the required protein domains of PYE needed for interaction with OLV. Different deletion constructs of PYE (**Fig 3A**) were tested in Y2H and in FRET-APB experiments. PYE harbors an ethylene-responsive element-binding factor-associated amphiphilic repression (EAR) motif in its C-terminal part. EAR motifs are commonly known as repression motifs in plants, commonly associated with TFs that function as adaptors to recruit TOPLESS and -related repressor proteins via EAR motif binding [25]. However, a function of the EAR motif in this regard may not be the case for the PYE protein [5]. Both, PYE and PYE-DEAR interacted with OLV in Y2H assays (**Fig 3B**). On the other hand, all other tested PYE deletion constructs, which were represented by the N-terminal part with bHLH domain (PYE-N), the C-terminal part without bHLH domain (PYE-C), PYE devoid of the bHLH domain (PYE-DbHLH) or only the bHLH domain (PYE-bHLH) did not show any interaction with OLV in Y2H assays (**Fig 3B**). Based on FRET-APB experiments, PYE-N, PYE-C and PYE-DbHLH can be suggested to interact with OLV, but the interaction strengths determined as FRET efficiencies were lower in the case of all three PYE mutants compared to PYE (**Fig 3C**). In one and two out of three experiments, FRET efficiency was lowest for PYE-C and PYE-DbHLH, respectively (**Fig 3C**).

Thus, PYE does not have a unique specific domain for interaction with OLV (**Fig 3D**). Clearly the EAR motif is not likely involved in the interaction. The Alphafold multimer tool was used to predict the structure of PYE-OLV proteins [26]. This model suggests how the three-dimensional structure of PYE can provide a binding site for OLV (**Fig 3E**). This prediction indicates an interaction interface of OLV near the PYE bHLH domain. An additional interface for OLV interaction is also predicted proximal to the C-terminal end of PYE (**Fig 3E**). Potentially, OLV may modulate the ability of PYE to control its target genes.

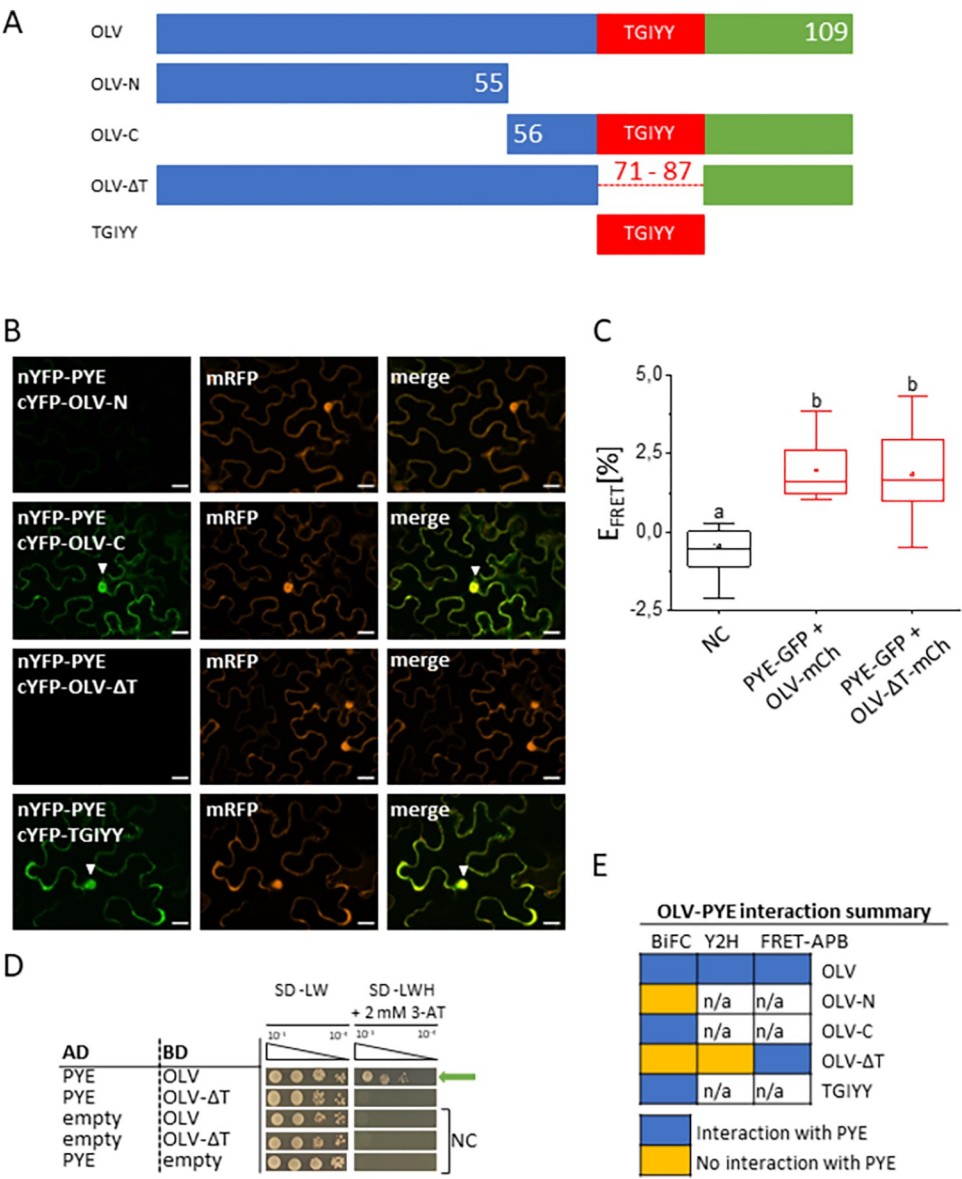

**Fig 2. The TGIYY motif of OLV can be important for the PYE-OLV interaction.** Schematic depiction of OLV-full-length (OLV-FL) and truncations. The positions of amino acid residues are indicated by numbers (compare with S1 Fig). The conserved TGIYY motif is depicted in red. Additional information in S2 to S4 Figs. **(B)** Bimolecular fluorescence complementation (BiFC) experiment to analyze interactions between cYFP-OLV truncations and N-YFP-PYE in transiently transformed tobacco leaf epidermis cells. mRFP was used as transformation control. YFP signals indicate protein interaction. Scale bars: 20 µm. Arrowheads indicate nuclear YFP signals. **(C)** Quantitative Förster resonance energy transfer-acceptor photobleaching (FRET-APB) measurements in transiently transformed tobacco leaf epidermis cells. Box plots indicate FRET efficiency ($E_{FRET}$) and strength of the interaction, determined in the nucleus using the pair PYE-GFP and OLV-mCherry (mCh) or OLV-DT-mC. NC: Donor-only PYE-GFP. An increased FRET efficiency ($E_{FRET}$) of the FRET pair compared to the NC is an indication for protein interaction. Different letters indicate statistically significant differences (one-way ANOVA and Tukey´s post-hoc test, $p < 0.05$), n = 15 nuclei. **(D)** Targeted yeast two hybrid (Y2H) assay between OLV and OLV-DT with PYE. Yeast co-transformed with activation domain (AD) and binding domain (BD) plasmid combinations were spotted in 10-fold dilution series ($A_{600} = 10^{-1}-10^{-4}$) on SD-LW (positive growth and co-transformation control) and SD-LWH + 2 mM 3-AT (selection for protein interaction) plates. Negative controls (NC): Empty AD with BD protein; AD protein with empty BD. **(E)** Summary of protein interactions from B-D. n/a, not analyzed.

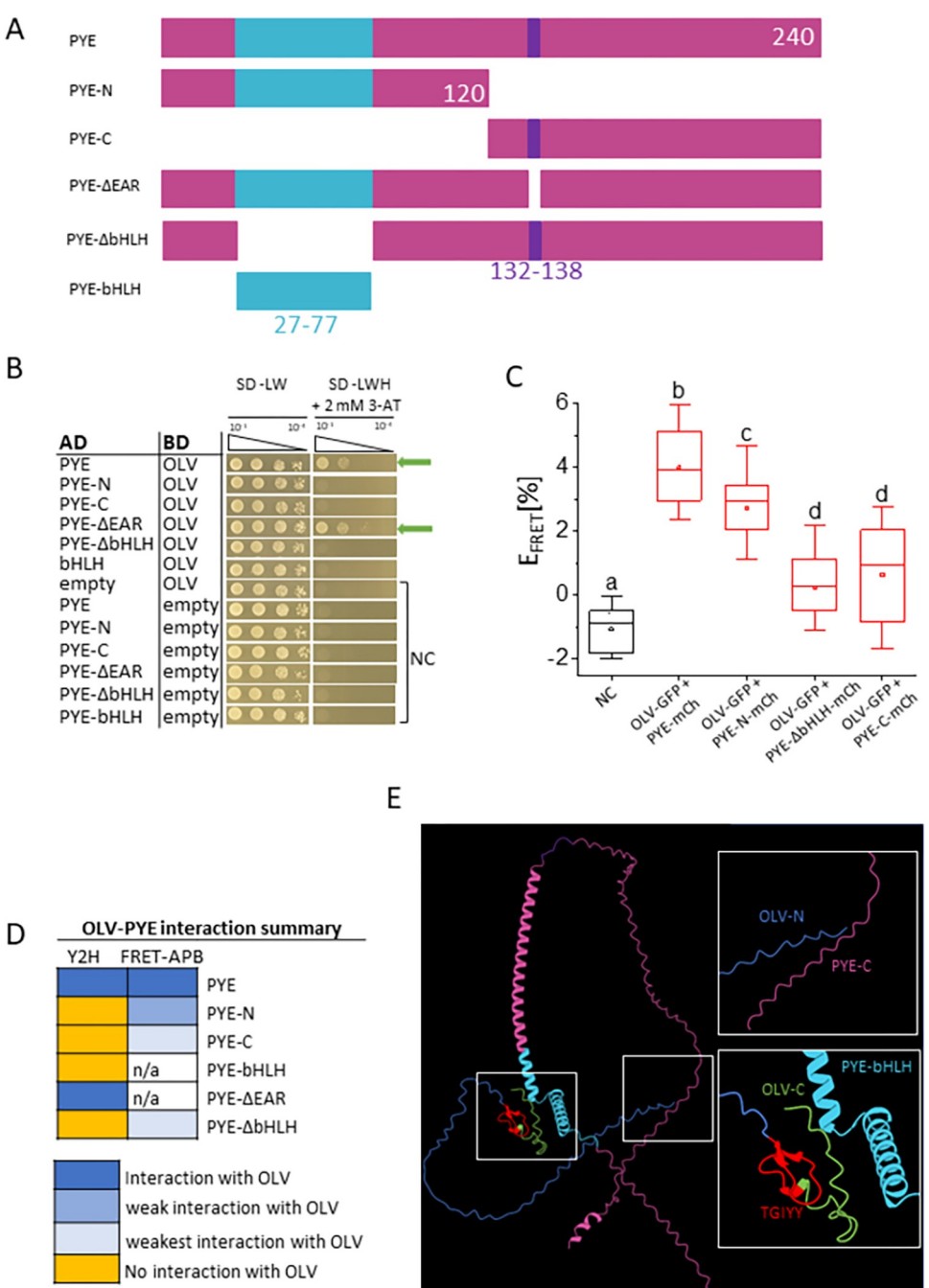

**Fig 3. PYE deletion mapping indicates that multiple regions of PYE are needed for the OLV-PYE protein complex. (A)** Schematic depiction of PYE-full-length (PYE-FL) and truncations used in B and C. The positions of amino acid residues are indicated by numbers. The conserved bHLH domain and EAR motif are depicted in cyan and violet. **(B)** Targeted yeast two hybrid (Y2H) assay between PYE-FL, PYE truncations and OLV. Yeast co-transformed with activation domain (AD) and binding domain (BD) plasmid combinations were spotted in 10-fold dilution series ($A_{600} = 10^{-1}-10^{-4}$) on SD-LW (positive growth and co-transformation control) and SD-LWH + 2 mM 3-AT (selection for protein interaction) plates. Negative controls (NC): Empty AD with BD protein; AD protein with empty BD. **C)** Quantitative Förster resonance energy transfer-acceptor photobleaching (FRET-APB) measurements in transiently transformed tobacco leaf epidermis cells. Box plots indicate FRET efficiency ($E_{FRET}$) and strength of the interaction, determined in the nucleus using the pair OLV-GFP and PYE-mCherry (mCh) or indicated PYE mutant-mCh. NC: Donor-only OLV-GFP. An increased FRET efficiency ($E_{FRET}$) of the FRET pair compared to the NC is an indication for protein interaction. Different letters indicate statistically significant differences (one-way ANOVA and Tukey´s

post-hoc test, p<0.05), n = 15 nuclei. (**D**) Summary of protein interactions from B, C. n/a, not analyzed. (**E**) Predicted structure of OLV-PYE proteins using Alphafold multimer tool (Evans et al., 2021). The color code indicates the protein domains investigated. Enlarged views are boxed.

Taken together, deletion mapping of the interaction interface of PYE corroborated the finding that OLV can target PYE.

## PYE and OLV colocalized in plant cells in the nucleus

A prerequisite for protein interaction in plants is that PYE and OLV proteins are located in the same cellular compartments and tissues.

The FRET-APB experiments described above already revealed that both proteins localize to the nucleus. More detailed protein localization and co-localization studies were performed to examine in which cell compartments PYE and OLV are localized when expressed alone or together, respectively. Single localization experiments indicated that PYE-mCherry localized primarily to the nucleus in *Nicotiana benthamiana* (tobacco) leaves, while weak signals were detected in the cytoplasm (**Fig 4A**). OLV-GFP, on the other hand, localized clearly to nucleus, cytoplasm and at strands of the endoplasmic reticulum (**Fig 4B**). When PYE-mCherry and OLV-GFP were co-expressed in tobacco leaves, they co-localized inside the nucleus (**Fig 4C**). The pattern did not change for OLVDT-GFP (**Fig 4B and 4C**). Therefore, co-localization studies suggest that OLV may interact with PYE primarily in the nucleus, whereby neither the presence of OLV nor that of the TGIYY motif changed the localization of PYE or OLV.

An interesting question was in which root tissues and root zones *OLV* and *PYE* are expressed. The promoter activities of both genes were assayed by using Arabidopsis lines that stably express the ß-glucuronidase (GUS)-encoding reporter gene driven by the *OLV* or *PYE* promoters. Detection of GUS activity is a sensitive method to determine promoter activities in different root zones. The lines were grown for six days under sufficient (+Fe) or deficient (–Fe) Fe supply and analyzed for promoter-GUS activity in roots (**Fig 4D–4G**). In the *proOLV*::*GUS* line, promoter driven-GUS activity was detected mainly in the cortex and epidermis of the root differentiation and root hair zones. GUS activity was not detected in the root tip and root elongation zone. GUS activity appeared patchy along the upper root zones (**Fig 4D and 4E**). *proOLV* lines grown under -Fe showed stronger GUS activity compared to plants grown under +Fe, implying an induction of the *OLV* promoter under -Fe conditions, as was expected. *ProPYE*::*GUS* expression had been reported previously for our growth conditions [11]. In accordance with these published data, strongest *PYE* promoter activity was detected in tissues of the inner stele, such as the pericycle, along the mature root and root hair zones, in all tissues at the root tip as well as of the root elongation and differentiation zones. In the root elongation and differentiation zones, *proPYE*-driven activity was detected in the cortex and epidermis (**Fig 4F and 4G**). Hence, *PYE* and *OLV* promoter activities overlapped in the root epidermis and cortex of the root differentiation zone. In other parts of the root, the promoter activities of the two genes indicated non-overlapping expression.

According to available transcriptome data under control conditions, *OLV* is expressed during germination and in the root epidermis [27,28].

In summary, PYE and OLV proteins can be present together in the root epidermis and cortex of the root differentiation zone, where the proteins may interact in the nucleus.

## OLV had a mild effect on PYE function

The next question was whether OLV affected the growth of plants or the activity of PYE. In our seedling growth assay, roots of Fe-deficient (- Fe) wild-type plants are longer than those of

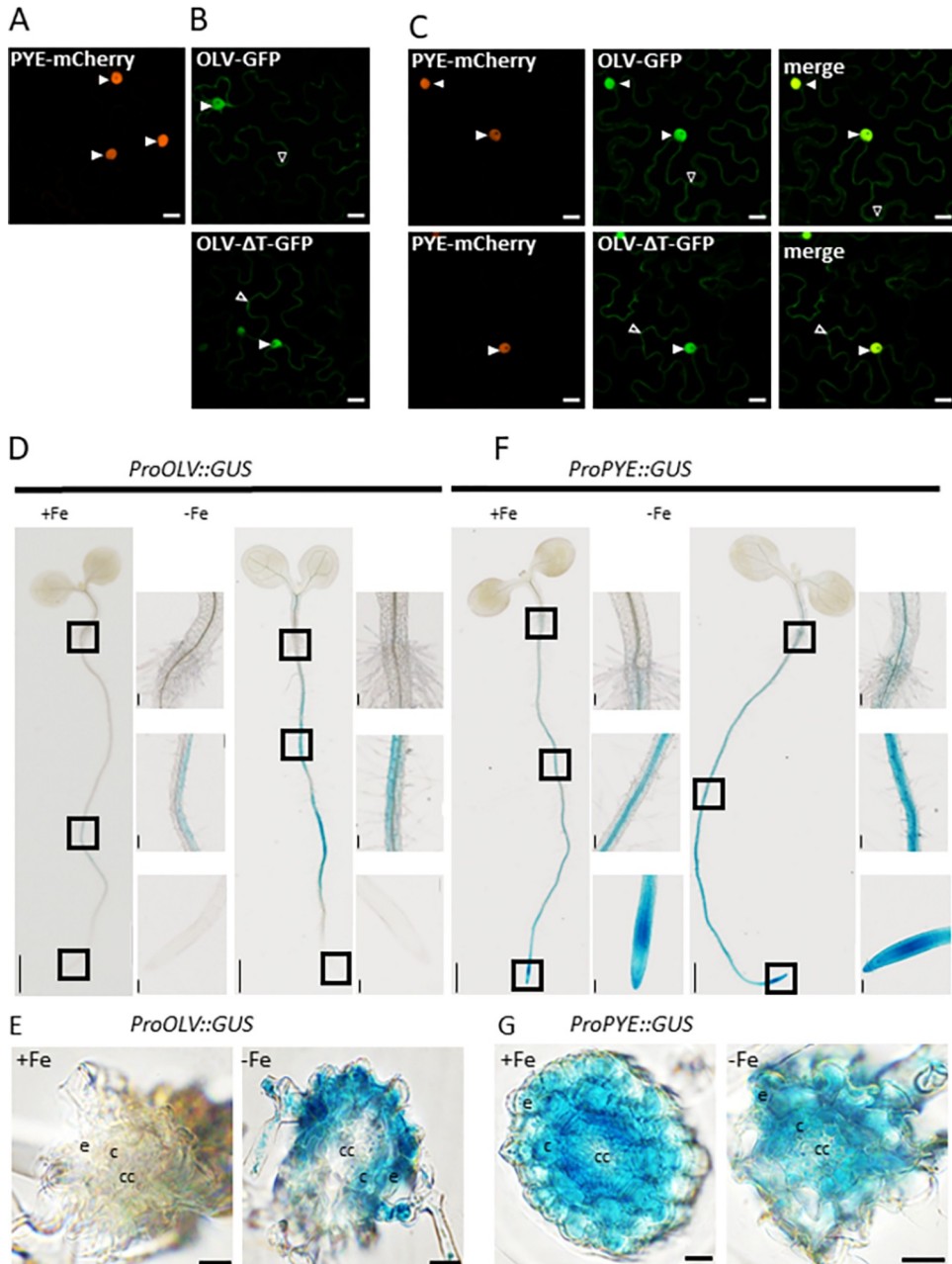

**Fig 4. PYE and OLV fluorescence fusion proteins colocalize in the nucleus and the *PYE* and *OLV* promoter activities overlap in the root differentiation zone.** (A-C) Subcellular localization of fluorescence fusion proteins in tobacco leaf epidermis cells upon **(A, B)**, single expression and **(C)**, co-expression. **(A)** PYE-mCherry; **(B)** OLV-GFP and OLV-ΔT-GFP, as indicated; **(C)** co-expressed PYE-mCherry / OLV-GFP and PYE-mCherry / OLV-ΔT-GFP, as indicated. Scale bars: 20 µm. Arrowheads indicate nuclear (filled arrowheads) and cytoplasmic (empty arrowheads) signals. OLV-ΔT is depicted in Fig 2A. **(D-G)** *OLV* and *PYE* promotor-driven beta-glucuronidase (GUS) reporter activity in Arabidopsis seedlings. Transgenic plants carrying either **(D, E)** pro*OLV*::*GUS* or **(F, G)** pro*PYE*::*GUS* were grown in the 6 d system under sufficient (+Fe) or deficient (-Fe) Fe supply and analyzed for GUS staining (blue precipitates). Rectangles in whole-seedling images (left side) indicate the area of enlarged images, from top to bottom: Root tip, root differentiation zone and transition from root to hypocotyl. Scale bar whole seedling images: 1 mm, close up images: 0.1 mm. **(E, G)** cross sections of the root differentiation zone from plants shown in **(D)** and **(F)**. Abbreviations; cc: Central cylinder, c: Cortex, e: Epidermis.

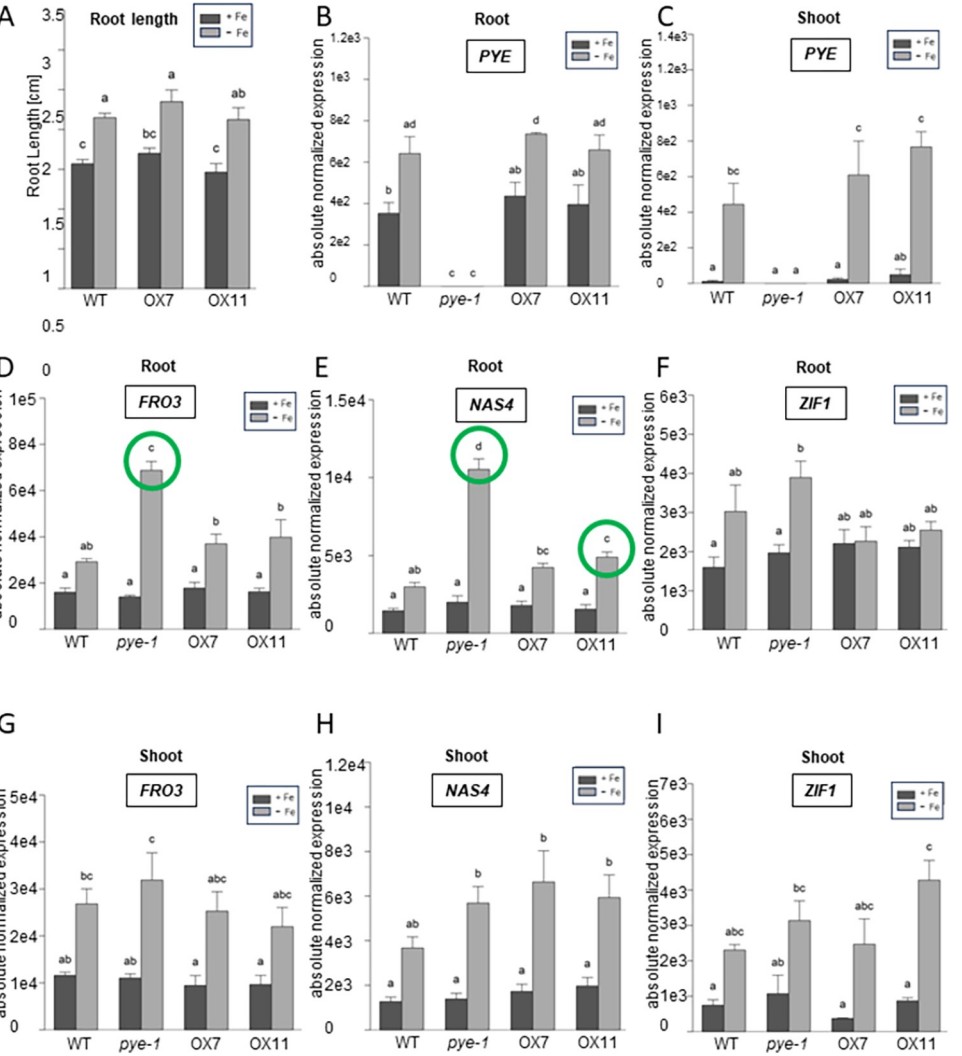

**Fig 5. Overexpression of OLV did not alter seedling root length and changed gene expression of PYE target genes only in one case (*NAS4*) in roots or shoots.** Two OLV overexpression lines were investigated (OX7, OX11; pro2x35S promoter, triple hemagglutinine-tagged HA₃-OLV) and compared with A-I, wild type (WT) and B-I, *pye-1*. Additional information in S1 File. **(A)** Root lengths of 7-day-old seedlings grown in + Fe or–Fe. **(B-I)** Gene expression analysis of *PYE*, *FRO3*, *NAS4* and *ZIF1* in root and shoot, as indicated in the figure. Plants were grown in the 9 + 3 d system with sufficient (+Fe) or deficient (-Fe) Fe supply for three days. The data are depicted as mean ± standard deviations; n = 3. Different letters indicate statistically significant differences (one-way ANOVA and Tukey´s post-hoc test, p<0.05). Green circles indicate significant differences to WT. Additional information in S2 and S3 Files, in S5 and S6 Figs.

Fe-sufficient (+ Fe) ones [29], and the root length may change in OLV mutant plants. Since PYE represses *FRO3*, *NAS4* and *ZIF1* (Long et al., 2010), we reasoned that a negative effect of OLV on PYE should result in higher expression levels, while a positive effect on PYE should cause lower expression levels of these PYE target genes.

At first, we assessed a possible impact of OLV overexpression (meaning expression behind the double cauliflower mosaic virus 35S promoter). We analyzed two hemagglutinine (HA₃)-tagged HA₃-OLV overexpressing lines (pro35S; OX7, OX11) (S5A–S5D Fig). HA₃-OLV protein (14.85 kDa) was detected in total protein extracts of 10 d-old seedlings independent of Fe supply, confirming the presence of HA-tagged OLV protein (S5B Fig). Root length was not changed upon overexpression of OLV (Fig 5A). *PYE* gene expression levels were elevated in–

Fe roots of wild type and OX7 and in–Fe shoots of wild type, OX7 and OX11 in comparison with the respective + Fe situation (**Fig 5B and 5C**). However, there were no significant differences in *PYE* levels between wild type and any OX line at—Fe or + Fe (**Fig 5B and 5C**). *FRO3* and *NAS4* were up-regulated in roots under–Fe versus + Fe in all tested lines and their expression was higher in *pye-1* roots versus wild type and versus Ox roots (**Fig 5D and 5E**). Interestingly, the only OX phenotype was a higher expression of *NAS4* compared to wild type, which was significant in the case of OX11 (**Fig 5E**). *ZIF1* was only up-regulated under—Fe versus + Fe in roots of *pye-1* versus wild type (**Fig 5F**). In shoots, there was a significant up-regulation of *FRO3*, *NAS4* and *ZIF1* at—Fe versus + Fe in the case of *pye-1* (*FRO3*, *NAS4*), OX7 (*NAS4*) or OX11 (*NAS4*, *ZIF1*), while there were no significant differences between the expression levels of any mutants with wild type (**Fig 5G–5I**). Since OLV Ox had a mild effect on PYE target genes, we tested whether OLV Ox altered the *pye-1* phenotype in any way. This was not the case, and *pye-1* OX7 as well as *pye-1* OX11 had the *pye-1* phenotype with respect to *FRO3* and *NAS4* up-regulation in roots (**S6 Fig**). Hence, we could first of all, reproduce the positive effect that *pye-1* has on the PYE target genes *FRO3* and *NAS4* in roots in our growth system, but we did not reproduce it neither in shoots nor for the reported target *ZIF1*. The only OLV overexpression phenotype observed was a positive effect on *NAS4* expression in roots, suggesting that OLV has a mild or partial effect on the repression of PYE in this case.

Next, we investigated *olv* loss-of-function mutant lines that we had generated by genome editing. Two alleles, *olv-3* and *olv-7*, had both premature stop codons in the *OLV* coding sequence (**S5E–S5G Fig**). When grown in the seedling assay, there was again no noticeable difference in the root lengths between wild type, *olv-3* and *olv-7* (**Fig 6A**). *PYE* gene expression was elevated at–versus + Fe in *olv-3* and *olv-7* roots or shoots, but not to a different level than in wild type (**Fig 6B and 6C**). In the *olv* mutants, PYE target genes *FRO3*, *NAS4* and *ZIF1* were similarly expressed as in wild type in roots and shoots (**Fig 6D–6I**), indicating that loss of OLV function had no effect on PYE action.

Finally, we studied whether OLV had any effect during later growth stages up to flowering. Interestingly, *olv-3* and *olv-7* had smaller rosettes in—Fe and + Fe-treated plants, while OLV OX7 and OX11 had no such phenotype (**Fig 7A–7D**). To study the *olv* phenotype in more detail, we investigated Fe and Zn contents. Remarkably, Fe contents were lower after three days of -Fe in the young leaves, but not old leaves of wild type, *olv-3* and *olv-7* (**Fig 7E and 7F**). Only *olv-7* had a statistically lower Fe content in the young leaves compared with wild type (**Fig 7E**). No differences were found for Zn (**Fig 7G and 7H**).

In summary, the only two phenotypes of *OLV* mutants detected were first, enhanced *NAS4* expression in OX roots, and second, reduced plant rosette growth of *olv* mutants at an advanced growth stage.

## Discussion

Small regulatory proteins are known that act as effectors to influence TF activities. We report here that a small protein, called OLV, can interact with the bHLH TF and Fe deficiency response regulator PYE. OLV alteration has an effect on expression of *NAS4*, a target gene of PYE in roots, and plant rosette growth. OLV orthologs are small proteins (mostly between 95 and 120 aa), that share the highly conserved TGIYY motif in their C-terminus. Due to their conservation in angiosperms, OLV may have a function associated with land plant evolution. The acquisition and mobilization of Fe are physiologically highly relevant in higher compared with lower land plants. Hence, small proteins may have become recruited to control Fe homeostasis regulators, such as bHLH TFs to fine-tune responses.

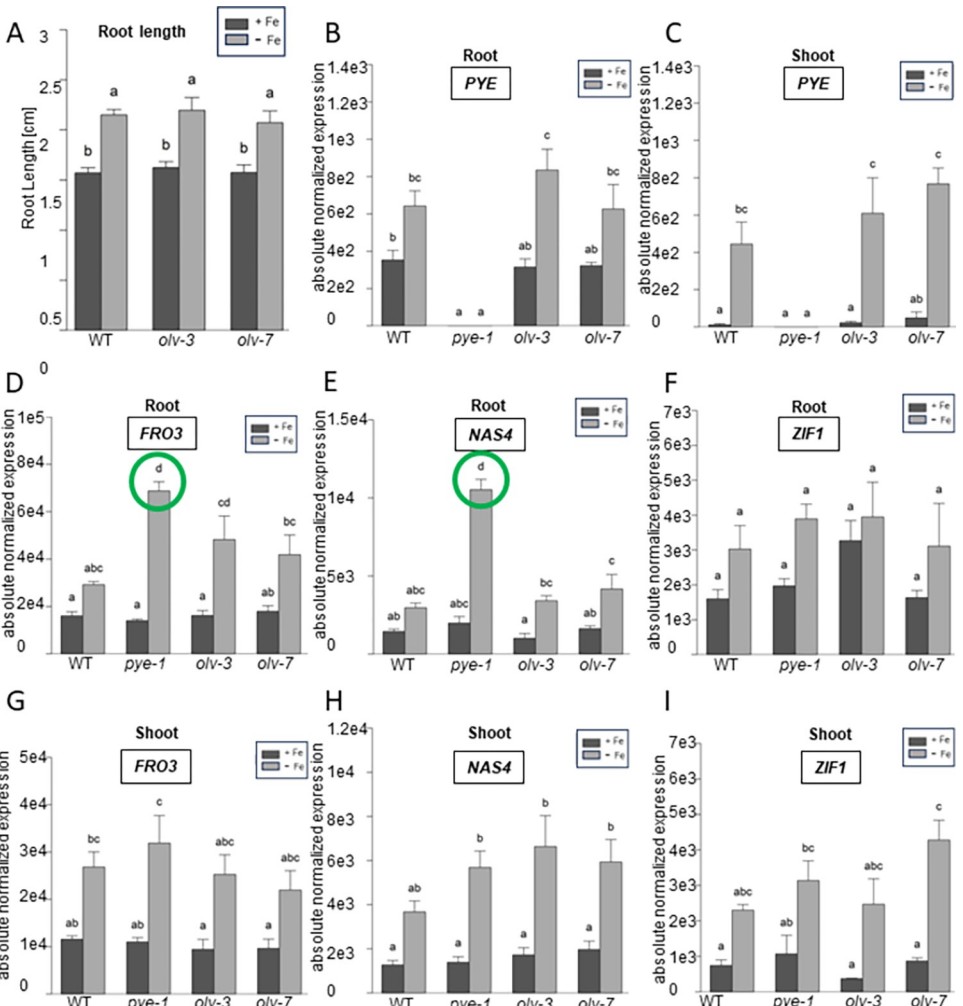

**Fig 6. *olv* loss of function did not alter seedling root length nor changed gene expression of PYE target genes significantly.** Two *olv* loss of function mutant lines, *olv-3* and *olv-7*, were investigated and compared with A-I, wild type (WT) and B-I, *pye-1* **(A)** Root lengths of 7-day-old seedlings grown in + Fe or–Fe. **(B-I)** Gene expression analysis of *PYE*, *FRO3*, *NAS4* and *ZIF1* in root and shoot, as indicated in the figure. Plants were grown in the 9 + 3 d system with sufficient (+Fe) or deficient (-Fe) Fe supply for three days. The data are depicted as mean ± standard deviations; n = 3. Different letters indicate statistically significant differences (one-way ANOVA and Tukey´s post-hoc test, p<0.05). Green circles indicate significant differences to WT. Additional information in **S2 and S3 Files** and in **S5 Fig**.

The interaction between PYE and OLV has been validated in independent approaches. First of all, protein interaction studies of three different kinds confirmed that OLV is able to bind PYE in different circumstances and cells. Second, these findings were corroborated by using deletion mutant constructs and mapping of the interaction face. Nevertheless, depending on the method, some of the deletion versions were or were not interacting. We therefore propose that the interaction interface is large between the two proteins, covering different domains and parts of the PYE and OLV protein, as was indicated in an AlphaFold structural prediction approach. Third, OLV and PYE interacted in the nucleus in BiFC and FRET-APB experiments. The two proteins also co-localized in the nucleus of tobacco epidermis cells when co-expressed. It is therefore very likely that the two proteins co-localize in nuclei of root differentiation zone epidermis and cortex where both *OLV* and *PYE* promoters are active. Fourth, the up-regulated expression of one of the two promising PYE targets, *NAS4*, in roots upon

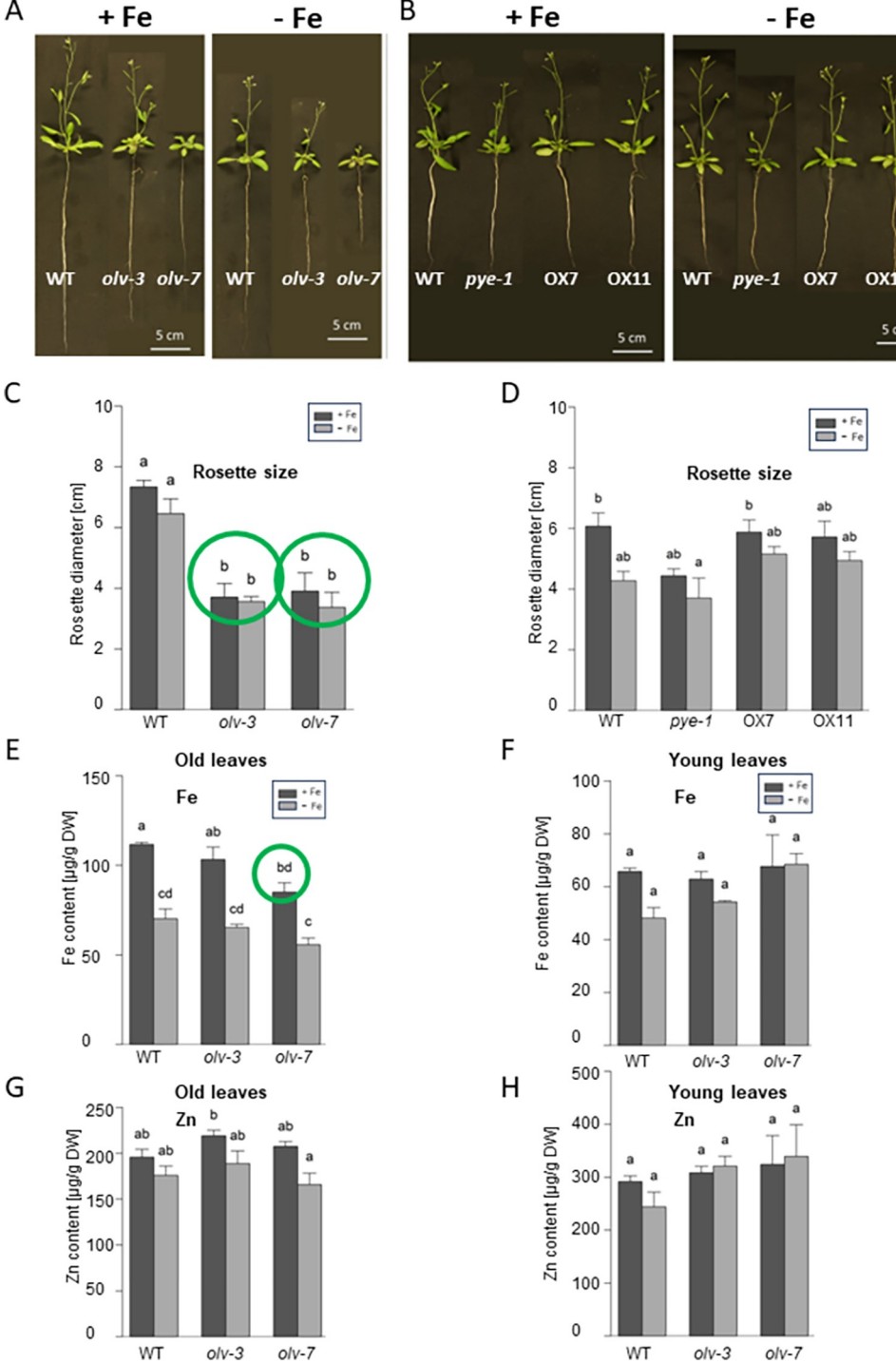

**Fig 7. *olv* loss of function mutants had a smaller rosette size and in one case lower Fe contents than wild type.** Two *olv* loss of function mutant lines, *olv-3* and *olv-7*, and two OLV overexpression lines were investigated (OX7, OX11; pro2x35S promoter, triple hemagglutinine-tagged HA₃-OLV) and compared with A-H, wild type (WT) and B, D, *pye-1*. Plants were grown in a hydroponic system and exposed to Fe sufficiency (+ Fe) and Fe deficiency (–Fe) for 3 days. Genotypes and growth conditions as indicated in the figure. **(A, B)** Photos of flowering plants grown in a hydroponic system. **(C, D)** Rosette size at bolting stage. **(E-H)** Metal contents per dry weight for **(E, F)** Fe and **(G, H)** Zn in **(E, G)**, young growing leaves and **(F, H)**, old and expanded leaves. The data are depicted as mean ± standard deviations; n = 3. Different letters indicate statistically significant differences (one-way ANOVA and Tukey´s post-hoc test, p<0.05). Green circles indicate significant differences to WT. Additional information in S4 to S9 Files.

OLV overexpression indicates that OLV may repress PYE function in the wild-type situation. This interpretation is consistent with the up-regulation of *NAS4* upon *pye-1* loss of function.

The conserved TGIYY motif is a characteristic of OLV. It was found to be relevant in protein interaction with PYE in Y2H and BiFC assays. Presence and absence of TGIYY did not affect the expression and localization of fluorescence fusion protein in tobacco. The presence of the TGIYY motif is evolutionarily more ancient than the evolving Fe deficiency bHLH regulatory cascade in angiosperms. Perhaps the TGIYY motif has a general role to allow protein interactions. Clearly, the TGIYY motif does not target the conserved EAR motif of the bHLH TF since there was no indication that the EAR motif is required for OLV interaction. Interestingly, the family of small IMA proteins which positively regulates the Fe deficiency response also has a characteristic conserved region in the C-terminus, while the N-terminus is variable [21,22]. The IMA conserved region is needed for protein interaction with co-expressed E3 ligases of the BTS/BTSL-type [11,12]. Hence, the co-expression and protein interaction represent interesting parallels between the IMA-BTS/L and the OLV-PYE systems. It can be proposed that Fe deficiency-induced small proteins harbor a variable region combined with a highly conserved C-terminal region for protein interaction with co-expressed target proteins and regulatory factors of the Fe deficiency cascade, important for the fine-tuning of individual responses.

*OLV* is mainly expressed in Fe-deficient seedling roots, while *PYE* is induced upon Fe deficiency in roots and shoots. In roots, *OLV* and *PYE* expression occurs in different regions, with the exception of an overlap in a small region in the outer tissue layers of the root differentiation zone. Despite of that, it has been reported that PYE is mobile across the root close to the root tip and has different functions in the tissue layers [4,30]. Perhaps the protein interaction of PYE with OLV is physiologically relevant in the root differentiation zone close to the root tip. In this region, OLV may affect *NAS4* gene expression, possibly through interacting with PYE. Interestingly, *NAS4* promoter is targeted by PYE but also by ILR3, and the PYE and ILR3 TFs interact [31]. Loss of function of ILR3 may also lead to an up-regulation of *NAS4* [31]. Perhaps OLV can affect the interaction between PYE and ILR3 or fine-tune the possible TF protein interaction complexes.

An interesting observation was that rosette growth was reduced in *olv* mutants compared with wild type. The reduced rosette growth can be a consequence of the decreased leaf cell elongation in old leaves of *olv* plants, which can be considered the source leaves for Fe [32]. Perhaps, there is a long-distance signaling between the action of OLV in roots and the amount of Fe that is translocated to shoots to allow for leaf expansion in these old leaves.

## Conclusion and perspectives

OLV is a small protein interactor of PYE that may interact via a conserved previously non-characterized TGIYY motif in the Fe deficiency response. OLV had a mild effect on plant growth and Fe deficiency responses involving *NAS4*. Since *NAS4* is specifically targeted by other TFs of the Fe deficiency response, OLV may control a particular sub-function of PYE in controlling *NAS4*. OLV is primarily expressed during germination and in in roots where it is induced by Fe deficiency. Future studies can address the functional and structural relevance of the TGIYY motif for formation of the protein complex with PYE. Protein interaction can be further studied in plant roots, e.g. by co-expressing fusion proteins of OLV and PYE expressed from their respective promoters.

## Materials and methods

### Relevant DNA sequence accessibility

Sequence data of PYE, OLV, AKT1 and CIPK23 can be found in the TAIR library with the following accession numbers: AKT1 (AT2G26650), BHLH39 (AT3G56980), CIPK23

(AT1G30270), FRO3 (AT1G23020), ILR3 (AT5G54680), NAS4 (AT1G56430), PYE (AT3G47640), OLV (AT1G73120), OPT3 (AT4G16370), ZIF1 (AT5G13740).

## Plant Material and generation of transgenic lines

The Arabidopsis ecotype Col-0 (Columbia-0) was used as wild type (WT) and background for all transgenic lines constructed in this study. *pye1-1* has been previously described (Long et al. 2010). To generate triple $HA_3$-tagged overexpression lines, full-length coding sequence (CDS) of *PYE* and *OLV* were amplified from cDNA of Fe deficient Arabidopsis WT roots and transferred via Gateway cloning into pDONR207 (Invitrogen) according to the manual (BP reaction, Thermo Fisher Scientific). After sequencing, the respective CDS was shuttled into the plant binary destination vector pAlligator2, via Gateway LR reaction (Thermo Fisher, Scientific). pAlligator2 allows ectopic overexpression of genes N-terminally tagged with a triple HA under the control of a double CaMV 35S promoter [33]. Final constructs were sequenced and subsequently transferred into agrobacteria (*Rhizobium radiobacter* strain GV3101 (pMP90) [34]. The Agrobacterium mediated floral dip method [35] was applied to generate stable transgenic Arabidopsis lines. Transformed seeds were selected based on seed-specific GFP expression and confirmed by genotyping PCR (**S1 Table**). Homozygous T3 plants were used for further analysis. To generate pro*OLV*::*GUS* the promoter sequence of *OLV* (988 bp upstream of start codon) was amplified from Arabidopsis WT leaf gDNA (**S1 Table**), transferred into pDONR207 (Invitrogen) via Gateway BP reaction (Thermo Fisher, Scientific) and then sequenced. To generate the final vector, the promoter sequence was shuttled into the Gateway binary destination vector pGWB3 (Nakagawa et al., 2007) via LR reaction (Thermo Fisher, Scientific), followed by sequencing. Stable Arabidopsis lines were constructed as described above. The pro*PYE*::*GUS* line was previously described by [11]. To obtain *olv* loss of function mutants, the genome sequence was edited at the *OLV* locus by Crispr/Cas9 system according to published procedures [36]. sgRNAs were designed using the CRISPR-P2.0 tool [37], targeting the region of the beginning of the second exon up to the start of the TGIYY motif-encoding region. Forward and reverse guide RNA constructs were generated (`attgTGAATTGAT TTCCTAAGCAT` and `aaacATGCTTAGGAAATCAATTCA` including restriction sites for later genotyping) and transferred under the control of the Arabidopsis U6-26S RNA polymerase III promotor into pFH6_new. Subsequently, the whole sgRNA cassette was transferred into pFH1 (pUB-CAS9) via Gibson cloning (Thermo Fisher Scientific). Following Rhizobacter transformation and floral dipping, as described above, hygromycine-resistant plant lines were selected in the T1 and multiplied up to the T3. Genomic PCR using primers OLV genotyping_F and OLV genotyping_R surrounding the genome-edited region combined with DNA sequencing (primers OLV sequencing_F and OLV sequencing_R) was used to select loss of function lines *olv-3* and *olv-7*. Tobacco (*Nicotiana benthamiana*) plants were used for transient expression experiments.

## Plant growth conditions

Arabidopsis seeds were surface-sterilized and stratified for two days. Propagation and seed production was performed on soil in a climate chamber under long day conditions (16 h light, 8 h dark, 21°C). Three different growth systems were used for phenotypic, physiological, and histochemical analyses. For plate growth systems, Arabidopsis seedlings were grown on upright sterile plates containing modified half-strength Hoagland medium [1.5 mM Ca $(NO_3)_2$, 1.25 mM $KNO_3$, 0.75 mM $MgSO_4$, 0.5 mM $KH_2PO_4$, 50 µM KCL, 50 µM $H_3BO_3$, 10 µM $MnSO_4$, 2 µM $ZnSO_4$, 1.5 µM $CuSO_4$, 0.075 µM $(NH_4)_6MO_7O_{24}$, 1% (w/v) sucrose, pH 5.8, containing 1.4% plant agar (Duchefa)] and supplemented with either sufficient Fe (50 µM

FeNaEDTA, +Fe) or no Fe (0 μM FeNaEDTA). After stratification, the plates were transferred into plant growth chambers (CLF Plant Climatics) with long day conditions for 7 days with +Fe or -Fe (7 d system: whole seedlings were analyzed for root lengths). Plants grown in the 9 + 3 d system were grown under +Fe for 9 days and subsequently transferred on +Fe or -Fe for another three days. Roots and shoots were harvested separately for gene expression analysis. For hydroponic growth, 12-day-old +Fe plate-grown seedlings were transferred to quarter-strength Hoagland hydroponic medium containing 25 μM FeNaEDTA as + Fe. The hydro-ponic medium was changed regularly and aerated. At day 30 of plant age, roots were washed three times and transferred to +Fe (25 μM Fe) and -Fe (0 Fe) for 3 days.

Tobacco plants were grown on soil for three to four weeks in a greenhouse facility under long day conditions. After tobacco leaf infiltration the plants were kept at room temperature under long day light conditions in the lab for two to three days until localization and protein interaction studies were performed.

## Construction of OLV and PYE protein mutant versions

pDONR207:OLV was used as a template to generate different OLV truncated versions. For OLV-N (aa 1 to 55), the primer pair OLV_B1 fw/OLV +165 bp _B2 rev was used, and for OLV-C (aa 56 to 109) the primer pair OLV + 165 bp_B1 fw/OLV_B2 rev. To amplify the con-served motif (OLV-TGIYY, aa 71 to 87) the primer pair OLV + 210 bp_B2 fw/OLV + 261 bp_B2 rev was used. Deletion of TGIYY (OLV-ΔTGIYY, aa 1–70 and 88–109) was generated by overlap-extension PCR. Two partially overlapping parts of OLV were amplified with the primer pair OLV_B1 fw/OLVΔ rev and OLVΔ fw/OLV_B2 rev (S1 Table). Both amplicons served as template in a second PCR to amplify OLVΔ with the underlined outer primers. All primers (except OLVΔ rev and OLVΔ fw) carry B1 and B2 Gateway attachment sites. Subse-quently all amplicons were transferred into pDONR207 (Invitrogen), sequenced, and shuttled into destination vectors.

pAlligator2:PYE was used as a template to generate different PYE truncated versions. For PYE-N (aa 1 to 120), the primer pair PYE_B1 fw/ PYE+360_attB2 rev was used, and for PYE-C (aa 121–240) the primer pair PYE+361_attB1 fw/PYE_B2 rev was used. To amplify bHLH domain (PYE-bHLH), aa 27 to 77 the primer pair PYE_attB1 fw +81bp/PYE_attB2 rev +231bp was used. Deletion of bHLH (PYE-ΔbHLH, aa 1–26 and 78–240) was generated by overlap-extension PCR. Two partially overlapping parts of PYE were amplified with the primer pairs PYE_B1 fw/PYE_ΔbHLH rev and PYE_ΔbHLH fw/PYE_B2 rev. Both amplicons served as template in a second PCR to amplify PYE-ΔbHLH with the underlined outer primers. Dele-tion of EAR motif (PYE-ΔEAR, aa 1–131 and 139–240) was generated by overlap-extension PCR. Two partially overlapping parts of PYE were amplified with the primer pairs PYE_B1 fw/ PYE_ΔEAR rev and PYE_ΔEAR fw/PYE_B2 rev. Both amplicons served as template in a sec-ond PCR to amplify PYE-ΔEAR with the underlined outer primers. All primers (except PYE_ΔbHLH rev, PYE_ΔbHLH fw, PYE_ΔEAR rev, PYE_ΔEAR fw) carry B1 and B2 Gateway attachment sites. Subsequently all amplicons were transferred into pDONR207 (Invitrogen), sequenced, and shuttled into destination vectors.

## Targeted Yeast Two-Hybrid (Y2H) assay

To study protein-protein interactions full-length OLV and mutant versions of OLV were tested with PYE in a targeted Y2H assay. All constructs were N-terminally fused to the GAL4-AD (vector: pACT2-GW), which acts as prey within the Y2H system (AD, activation domain). To generate the bait, all constructs were N-terminally fused to the GAL4-BD (vector: pGBKT7-GW) (kindly provided by Dr. Yves Jacob, Institut Pasteur, Paris, France) (BD,

binding domain). The CDS of PYE and OLV were amplified from cDNA of Fe deficient Arabidopsis WT roots with primer pairs carrying B1 and B2 attachment sites (S1 Table) and transferred via Gateway cloning into pDONR207 (Invitrogen) according to the manual (BP reaction, Thermo Fisher Scientific). pDONR207 constructs were sequenced, the respective CDS shuttled into the destination vectors pACT2-GW/pGBKT7-GW via Gateway LR reaction (Thermo Fisher, Scientific) followed by additional sequencing. Bait and prey constructs were co-transformed into the yeast strain AH109 via the lithium acetate (LiAc)/SS carrier DNA/PEG method based on [38]. Briefly, a 50 ml AH109-YPDA culture was grown up to $OD_{600}$ = 0.5 and then made competent by the addition of 100 mM LiAc. 50 µl competent yeast cells were mixed with 33.3% (w/v) PEG 4000, 0.1 M LiAc, 50 µg denatured Calf Thymus DNA (Invitrogen), 0.5–0.7 µg AD-plasmid, 0.5–0.7 µg BD-plasmid and sterile water to a final volume of 360 µl for each transformation event. Heat shock treatment was performed at 42˚C for 20 min. Yeast cells were cultivated on minimal synthetic defined (SD) media (Clontech), lacking Leu (selection for pACT2-GW) and Trp (selection for pGBKT7-GW) for 2–3 days at 30˚C, to select for positive double transformants. As negative controls, bait or prey were combined with empty BD or AD plasmids and used in Y2H assays. As positive control the combination of pGBT9.BS:CIPK23 and pGAD.GH:cAKT1 was used (Xu et al., 2006). To test for protein interactions, overnight liquid cultures of transformed AH109 were adjusted to $OD_{600}$ = 1, and ten-fold serial dilutions down to $10^{-4}$ in sterile water were prepared. 10 µl of each suspension were spotted on SD agar plates lacking Leu, Trp and His and supplemented with 0.5 mM 3-amino-1,2,4-triazole (SD-LWH + 3-AT, suppression of background growth, detection of interaction). In parallel 10 µl of the same serial dilutions were spotted on SD agar plates lacking Leu and Trp (SD-LW, positive growth and double transformation control). Plates were cultivated at 30˚C for 7 d. Growth was documented by photographing the plates every second day. Final pictures were taken on day 7.

## Subcellular (co-) localization

Subcellular protein localization studies were performed to analyse the localization of proteins. Therefore, fluorophore tagged proteins were transiently expressed in tobacco leaf epidermis cells. For N-terminal fusions the pDONR entry clones with PYE and OLV sequences were used for Gateway LR reaction (Gateway, Thermofisher, Scientific) to generate the destination vectors based on pH7WGY2 (N-terminal YFP) (Karimi et al., 2005). Final constructs were sequenced. For C-terminal fusions the CDS of PYE and OLV versions were amplified from cDNA of Fe deficient Arabidopsis WT roots with primer pairs PYE_B1 fw/PYEns_B2 rev and OLV_B1 fw/OLVns_B2 rev carrying B1 and B2 attachment sites without stop codon or from existing pDONR with N-terminal fusion (S1 Table) and transferred via Gateway cloning into pDONR207 (Invitrogen) according to the manual (Gateway, BP reaction, Thermo Fisher Scientific). After sequencing, the respective CDS was shuttled into the destination vectors pMDC83 (C-terminal GFP fusion) [39], as well as into the ß-estradiol-inducible pABind-GFP/pABind-mCherry (C-terminal GFP and mCherry fusions) [40], via Gateway LR reaction (Thermo Fisher, Scientific). Constructs were sequenced and transformed into Agrobacteria strain GV3101 (pMP90). For tobacco leaf infiltration, an overnight culture of Agrobacteria, carrying one of the constructs, was centrifuged and the pellet re-suspended in AS medium (250 µM acetosyringone (in DMSO), 5% (w/v) sucrose, 0.01% (v/v) silwet, 0.01% (w/v) glucose), according to Bleckmann et al., 2010. The suspension was adjusted to $OD_{600}$ = 0.4 and infiltrated with a 1 ml syringe into the abaxial side of two tobacco leaves on two different plants. For co-localization experiments using GFP- and mCherry-tagged proteins, both Agrobacteria suspensions were mixed 1:1 to obtain a final $OD_{600}$ = 0.4 for each. Transformed

tobacco plants were kept in the lab for 48 to 72 h at RT under long day conditions (16 h light, 8 h dark). To induce the expression of pABind constructs, infiltrated tobacco leaves were sprayed abaxially with 20 μM ß-estradiol (in DMSO, supplemented with 0.1% (v/v) Tween20) 16 h before imaging. To analyze transgene expression and protein localization, 0.5 cm leaf discs were punched out and imaged using alaser-scanning confocal microscope (LSM780, Zeiss). GFP and YFP were imaged at an excitation wavelength of 488 nm and emission wavelength of 491 to 533 nm. mCherry was imaged at an excitation wavelength of 561 nm and emission wavelength of 562 to 626 nm. Localization and co-localization experiments were performed in three independent experiments with two infiltrated leaves of two different plants.

## Bimolecular Fluorescence Complementation (BiFC)

To verify PYE interactions with OLV *in planta* the BiFC 2in1 vector system was applied (Grefen and Blatt, 2012). The CDS of PYE and OLV versions was amplified from pDONR207 constructs (see "Plant Material") or cDNA of Fe deficient Arabidopsis WT roots using primer pairs carrying B3, B2 and primer pairs carrying B1, B4 attachment sites (S1 Table). Via BP reaction (Gateway, Thermo Fisher) all amplicons carrying B3, B2 attachment sites were transferred into pDONR221-P3P2 (Invitrogen, for nYFP fusion) and amplicons carrying B1, B4 attachment sites into pDONR221-P1P4 (Invitrogen, for cYFP fusion). Constructs were sequenced. OLV-FL or one of the truncated OLV versions were shuttled simultaneously with PYE into the destination vector pBiFC-2in1-NN [24] (N-terminal nYFP and cYFP fusions) via multisite LR reaction (Gateway, Thermo Fisher). Hereby pBiFC-2in1-NN:OLV-FL-PYE and pBiFC-2in1-NN:PYE-OLV-FL (additionally all truncated OLV versions were cloned into pBiFC-2in1-NN combined with PYE as described for OLV-FL) were created and sequenced. An internal mRFP, served as transformation control. As negative controls for PYE and OLV proteins that do not interact with either partners were selected. For PYE ILR3 was used as negative control. As negative control of OLV bHLH39 was chosen. Therefore, pBiFC-2in1-NN: PYE-bHLH39, pBiFC-2in1-NN:bHLH39-PYE, pBiFC-2in1-NN:OLV-ILR3 and pBiFC-2in1-NN:ILR3-OLV were cloned as described above. All constructs were transformed into Agrobacteria strain GV3101 (pMP90) and used for tobacco leaf infiltration as described in "Subcellular (co-) localization". After 48 to 72 h, cells which were mRFP positive were analysed for YFP signals using the Axio Imager M2 (Zeiss) with ApoTome. mRFP was imaged at an excitation wavelength of 545 nm and emission wavelength of 570 to 640 nm, YFP was imaged at an excitation wavelength of 524 nm and emission wavelength of 520 to 550 nm. Three independent BiFC experiments were performed, using two leaves for each construct.

## Förster Resonance Energy Transfer After Photo Bleaching assay (FRET-APB) between PYE and OLV

To investigate protein-protein interaction between PYE and OLV via FRET-APB, the full-length CDS of PYE and OLV were cloned into pABind-GFP, pABind-mCherry and pABind-GFP-mCherry (pABindFRET) [40] as described above. For FRET-APB experiments, tobacco leaves were infiltrated with Agrobacteria carrying pABind-GFP:PYE and pABind-mCherry: OLV, or vice versa, to determine the strength of the protein-interaction ability. GFP-tagged proteins with donor only (pABind-GFP:PYE or OLV) served as negative control, the corresponding protein fused to a double tag of GFP-mCherry (pABindFRET:PYE or OLV) as positive control. To induce gene expression, infiltrated tobacco leaves were sprayed with 20 μM ß-estradiol 24 h after infiltration. The experiment was performed 20 h after ß-estradiol treatment.

FRET-APB measurements were taken with a laser-scanning confocal microscope (LSM 780, Zeiss) and controlled by the ZEN2 Black Edition software (Zeiss). For both fluorophores the fluorescence intensity was determined in the nucleus. GFP was imaged at an excitation wavelength of 488 nm and emission wavelength of 491 to 533 nm. mCherry was imaged at an excitation wavelength of 561 nm and emission wavelength of 562 to 626 nm. The measurement parameters were 20 frames of a 128 x 128 pixel format with a pixel dwell time of 2.55 μs. mCherry was photobleached after the 5th frame, using 100% laser power at 561 nm and 80 iterations. The FRET efficiency (FRET E) was calculated in percent of the relative GFP intensity increase after mCherry acceptor photobleachingwith the following equation FRET E [%] = $[(I_{D\ after}—I_{D\ before}) / I_{D\ after}] * 100$ ($I_{D\ after}$ = donor intensity after acceptor photobleaching, $I_{D\ before}$ = donor intensity before acceptor photobleaching). Three independent experiments analyzing at least 15 nuclei with equal expression of both fluorophores were performed.

Original FRET-APB imaging data sets are available at https://www.ebi.ac.uk/biostudies/bioimages/studies/S-BIAD1028 with the accession number S-BIAD1028.

## Multiple sequence alignmet of OLV homologues

A BLAST search of Arabidopsis OLV-FL aa as well as the TGIYY motif sequence was performed in every order of the angiosperms, selected lower plants and other non-plant organisms [41] using NCBI blastp. The protein sequence of the hit with the highest maximum score of each order was re-blasted in another BLAST analysis against the Arabidopsis TAIR10 protein sequence collection, applying TAIR BLAST 2.2.8 for validation. Multiple sequence alignments of all members with highest maximum score of each order were performed using the Clustal Omega algorithm [42] and visualized with Jalview 2.10.4 [43].

## Histochemical ß-glucuronidase (GUS) assay

Pro*PYE*:*GUS* transgenic plants have previously been reported [11]. Pro*OLV*:*GUS* lines have been generated as described above. Two independent pro*OLV*:*GUS* and pro*PYE*:*GUS* lines were chosen and propagated to T2 or T3 for further analysis. Plants were grown in the 6 d system on +Fe and -Fe, in two biological replicates, and assayed for histochemical GUS activity. Four to six seedlings of each line were incubated in 1 ml GUS staining solution containing [50 mM $Na_2HPO_4$, 50 mM $NaH_2PO_4$ pH 7.2, 2 mM $K_4[Fe(CN)_6]Fe^{2+}$, 2 mM $K_3[Fe(CN)_6]Fe^{3+}$, 0.2% Triton-X, 2 mM 5-bromo-4-chloro-3-indoyl-b-D-glucuronic acid (X-Gluc)] [44] for 15 min to four hours at 37˚C in the dark, until blue staining was observed. Afterwards leaf chlorophyll was removed by incubation in 70% EtOH for 24 h (EtOH was exchanged every few hours). Whole seedlings were imaged with the Axio Imager M2 (Zeiss, 10x objective magnification). Single images were assembled by using the stitching function of the ZEN 2 BLUE Edition (Zeiss).

## Root length measurement

Plants were grown on Hoagland agar plates (see "Plant Growth Conditions") for 7 d on + Fe and -Fe. Seedlings were imaged on day 6. Primary root length of individual seedlings was measured using the JMicroVision software (Version 1.2.7, https://jmicrovision.github.io/v127/install127.htm). Root length was measured in two independently grown sets of plants with 45 to 60 plants for each line and condition.

## Gene expression analysis by RT-qPCR

Gene expression analysis was performed as described [45]. In brief, total RNA was either extracted from whole seedlings grown in the 6 d system (n = 60–70 plants per replicate) or from roots/shoots of plants grown in the 9 + 3 d system (n = 20–25 roots/shoots per replicate), using the peqGOLD Plant RNA KIT (PeqLab). Reverse transcription using oligo(dt) primer and the RevertAid first-strand cDNA synthesis kit (Thermo Fisher, Scientific) was performed to obtain cDNA. RT-qPCR was carried out on the SFX96 Touch$^{TM}$ Real Time Detection System (Bio-Rad) with the iTag$^{TM}$ Universal SYBR® Green Supermix (Bio-Rad) according to the manual. The Bio-Rad SFX Manager$^{TM}$ software (version 3.1) was applied to process the data. Absolute gene expression values were calculated by gene specific mass standard curve analysis. Data was normalized to the Arabidopsis elongation factor EF1B$\alpha$. All primer pairs for this study are listed in **S1 Table**. The experiment was performed with at least three biological and two technical replicates.

## Statistical analysis

For statistical analysis a one-way analysis of variance (ANOVA) and a Tukey´s post-hoc test, which allow the comparison of more than two groups, were performed. Null hypothesis was rejected for p-values smaller than 0.05. Different letters indicate significant differences ($p < 0.05$).

## Immunoblot analysis

Total proteins were extracted from ground plant material of either whole Arabidopsis seedlings grown in the 10 d system (Arabidopsis seedlings: n = 40) or from tobacco leaves. Protein extraction, SDS-PAGE and immunodetection was performed as previously described in (Le et al., 2016). In summary, frozen plant material was homogenized using the Precellys 24 (Peqlab Life Science, VWR) and proteins were extracted with 2x SDG buffer (62 mM Tris-HCL pH 8.6, 2.5% (w/v) SDS, 2% (w/v) dithiothreitol, 10% (w/v) glycerol, 0.002% (w/v) bromphenol blue). Samples containing equal amounts of protein were separated on 12% (w/v) SDS polyacrylamide gels via electrophoresis, followed by the protein transfer to a Protran nitrocellulose membrane (GE Healthcare). To control for equal loading of whole protein, proteins on the membrane were stained using PonceauS (0.2% (w/v) PonceauS, 3% (w/v) trichloroacetic acid, 3% (w/v) sulfosalicylic acid). To detect HA-tagged PYE or OLV protein the membranes were blocked in 5% (v/w) milk solution (Roth) in 1x TBST (20 mM Tris-HCL pH 7.4, 180 mM NaCl, 0.1 mM (v/v) Tween20) for 30 min to avoid nonspecific antibody binding. Afterwards, the membranes were incubated with anti-HA-peroxidase high-affinity monoclonal rat antibody (clone 3F10; Roche) 1:1000 diluted in 2.5% (w/v) milk-TBST solution, followed by three times washing in 1x TBST for 15 min each. To detect chemiluminescence signals of HA-tagged proteins the FluorChemQ System for quantitative western blot imaging (ProteinSimple) was applied and images were processed by the AlphaView® software (version 3.4.0.0, ProteinSimple).

## Supporting information

**S1 Fig. Co-expression of *PYE* and *OLV* genes.** Co-expression network of Fe deficiency-responsive genes. *OLV* belongs to a sub-network of FIT target genes (surrounded by a blue dashed line). FIT is a transcription factor required for up-regulated expression of FIT target genes in response to Fe deficiency in roots of seedlings (Schwarz and Bauer, 2020). *PYE* belongs to a sub-network of Fe homeostasis genes (surrounded by a red dashed line) up-

regulated upon Fe deficiency in roots and shoots of seedlings, also in the absence of FIT (Schwarz and Bauer, 2020). The ATTED-II tool (Ver. 9.2) was used to generate the network (Obayashi et al., 2018) based on *PYE* and *OLV* as input genes. The violet arrow links the two genes encoding the interacting proteins PYE and OLV.
(PDF)

**S2 Fig. OLV full-length protein sequence alignments.** (A) Multiple sequence alignment of OLV full length protein sequences from angiosperms. The ortholog with the highest maximum similarity score of each order is shown. *Arabidopsis thaliana* OLV is boxed in red. Conserved amino acid residues are shadowed in color. Below is the consensus sequence, with indicated TGIYY motif. See S3 Fig. (B) Sequence alignment of the OLV TGIYY motifs from *A. thaliana* and rice, highlighting a consensus.
(PDF)

**S3 Fig. Multiple sequence alignment of the TGIYY motifs from OLV orthologs of angiosperms and of TGIYY-containing proteins found in other organisms.** (A) Schematic representation of full length OLV and its amino acid sequence with N-terminus in blue, conserved TGIYY motif in red, C-terminus in green. (B) Multiple sequence alignment of the OLV TGIYY motifs of angiosperms. The ortholog with the highest maximum score similarity of each order is given. Multiple sequence alignments and consensus depicting TGIYY motifs from proteins found in (C), selected green algae and non-angiosperm land plants, and (D), non-plant organisms.
(PDF)

**S4 Fig. Localization of various YFP-OLV fusions.** Subcellular localization in tobacco leaf epidermis cells. Proteins were N-terminally tagged to YFP. The OLV truncations are described in Fig 2A. Scale bar: 20 μm.
(PDF)

**S5 Fig. OLV overexpression and *olv* mutant lines.** (A-D), Two OLV overexpression lines were investigated (OX7, OX11; pro2x35S promoter, triple hemagglutinine-tagged HA$_3$-OLV). (A), Schematic overview of transgenic T-DNA construct, derived from pAlligator plasmid, not drawn to scale. (B) Anti-HA immunodetection of HA$_3$-OLV protein in OX7 and OX11 grown under Fe-sufficient (+ Fe) and Fe-deficient (- Fe) conditions. Protein extracts were obtained from 10-d-old whole seedlings. PonceauS staining of the membrane served as loading control. Expected molecular weight of HA$_3$-OLV: 14.85 kDa. Additional information in **S1 File**. (C, D) Gene expression analysis of *OLV* in (C) root and (D) shoot OX plants, WT and *pye-1* mutant plants, as indicated in the figure. Plants were grown in the 9 + 3 d system with sufficient (+Fe) or deficient (-Fe) Fe supply for three days. (E-G), Two *olv* loss of function mutant lines, *olv-3* and *olv-7*, were investigated. (E), Schematic representation of full length OLV and its amino acid sequence with N-terminus in blue, conserved TGIYY motif in red, C-terminus in green, comparison with two mutant OLV protein versions resulting from deletion/insertion following a genome editing procedure in *olv-3* and *olv-7*. (F, G) Gene expression analysis of *OLV* in (F) root and (G) shoot of *olv* mutant, WT and *pye-1* mutant plants, as indicated in the figure. Plants were grown as in (C, D). The data in (C, D, F, G) are depicted as mean ± standard deviations; n = 3. Different letters indicate statistically significant differences (one-way ANOVA and Tukey´s post-hoc test, p<0.05). Additional information in **S3 File**.
(PDF)

**S6 Fig. Overexpression of OLV in the background of *pye-1* did not alter the *pye-1* phenotype.** Two OLV overexpression lines with *pye-1* background were investigated (OX7, OX11;

pro2x35S promoter, triple hemagglutinine-tagged HA$_3$-OLV) and compared with wild type (WT) and *pye-1*. (A) Root lengths of 7-day-old seedlings grown in sufficient (+ Fe) or deficient (–Fe) Fe supply. (B-I) Gene expression analysis of *PYE*, *FRO3*, *NAS4* and *ZIF1* in root and shoot, as indicated in the figure. Plants were grown in the 9 + 3 d system with three-day + and–Fe treatments. The data are depicted as mean ± standard deviations; n = 3. Different letters indicate statistically significant differences (one-way ANOVA and Tukey´s post-hoc test, p<0.05). Green circles indicate significant differences to WT. Additional information in S2 and S3 Files.
(PDF)

**S1 Table. Primers used in this study.**
(DOCX)

**S1 File. Original blot images.**
(PDF)

**S2 File. Root lengths 7-day seedlings Figs 5A, 6A and S6A.** Original data.
(TXT)

**S3 File. Gene expression values Figs 5B–5I and 6B–6I and S5C, S5D, S5F, S5G and S6B–S6I.** Original data.
(XLSX)

**S4 File. Rosette size Fig 7C.** Original data.
(TXT)

**S5 File. Rosette size Fig 7D.** Original data.
(TXT)

**S6 File. Fe contents young leaves Fig 7E.** Original data.
(TXT)

**S7 File. Fe contents old leaves Fig 7F.** Original data.
(TXT)

**S8 File. Zn contents young leaves Fig 7G.** Original data.
(TXT)

**S9 File. Zn contents old leaves Fig 7H.** Original data.
(TXT)

## Acknowledgments

We thank Gintaute Matthäi, Monique Eutebach, and Elke Wieneke for technical support. We thank Christopher Endres for technical help with *A. thaliana* root cross sections and Florian Hahn for guidance of CRISPR/Cas9 cloning. The authors thank Christopher Grefen, Bochum, and Andreas Weber, Düsseldorf, for providing plasmids. D.L., K.T., and B.S. were members of the international graduate school iGRAD-Plant/NEXT*plant*, Düsseldorf. D.B. is member of the international graduate school NEXT*plant*, Düsseldorf.

## Author Contributions

**Conceptualization:** Daniela M. Lichtblau, Mather Khan, Ksenia Trofimov.

**Formal analysis:** Daniela M. Lichtblau.

**Funding acquisition:** Petra Bauer.

**Investigation:** Daniela M. Lichtblau, Dibin Baby, Mather Khan, Ksenia Trofimov, Yunus Ari, Birte Schwarz.

**Methodology:** Daniela M. Lichtblau, Dibin Baby, Mather Khan, Ksenia Trofimov.

**Supervision:** Petra Bauer.

**Writing – original draft:** Daniela M. Lichtblau.

**Writing – review & editing:** Petra Bauer.

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
