## [Decision Letter · Decision Letter 0]

18 Jan 2024

PONE-D-23-39597The small iron-deficiency-induced protein OLIVIA and its relation to the bHLH transcription factor POPEYEPLOS ONE

Dear Dr. Bauer,

Thank you for submitting your manuscript to PLOS ONE. After careful consideration, we feel that it has merit but does not fully meet PLOS ONE’s publication criteria as it currently stands. Therefore, we invite you to submit a revised version of the manuscript that addresses the points raised during the review process.

Dear Authors,Please check the Reviewers' comments carefully and revise your work accordingly.Thank you very much for submitting an interesting work to PLOS ONE.Best regards,Nguyen 

We look forward to receiving your revised manuscript.

Kind regards,

Nguyen Hoai Nguyen

Academic Editor

PLOS ONE

 [This work was funded by Deutsche Forschungsgemeinschaft (DFG, German Research Foundation) under GRK F020512056 (NEXTplant) and Germany´s Excellence Strategy – EXC-2048/1 – project ID 390686111. Funding for instrumentation: Zeiss LSM780 + 4-channel FLIM extension (Picoquant): DFG- INST 208/551-1 FUGG.].  

[We thank Gintaute Matthäi, Monique Eutebach, and Elke Wieneke for technical support. We thank Christopher Endres for technical help with A. thaliana root cross sections and Florian Hahn for guidance of CRISPR/Cas9 cloning. The authors thank Christopher Grefen, Bochum, and Andreas Weber, Düsseldorf, for providing plasmids. D.L., K.T., and B.S. were members of the international graduate school iGRAD-Plant/NEXTplant, Düsseldorf. D.B. is member of the international graduate school NEXTplant, Düsseldorf. This work was funded by Deutsche Forschungsgemeinschaft (DFG, German Research Foundation) under GRK F020512056 (NEXTplant) and Germany´s Excellence Strategy – EXC-2048/1 – project ID 390686111. Funding for instrumentation: Zeiss LSM780 + 4-channel FLIM extension (Picoquant): DFG- INST 208/551-1 FUGG]

 [This work was funded by Deutsche Forschungsgemeinschaft (DFG, German Research Foundation) under GRK F020512056 (NEXTplant) and Germany´s Excellence Strategy – EXC-2048/1 – project ID 390686111. Funding for instrumentation: Zeiss LSM780 + 4-channel FLIM extension (Picoquant): DFG- INST 208/551-1 FUGG.].  

5. In the online submission form, you indicated that [Data will be published. Seeds and original data can be made available upon request.]. 

7. We note that Figure(s) 1A, 1C, 2B, 2D, 3B, 4A,B, C, D, E. 7A, 7B and S4 in your submission contain copyrighted images. All PLOS content is published under the Creative Commons Attribution License (CC BY 4.0), which means that the manuscript, images, and Supporting Information files will be freely available online, and any third party is permitted to access, download, copy, distribute, and use these materials in any way, even commercially, with proper attribution. For more information, see our copyright guidelines: http://journals.plos.org/plosone/s/licenses-and-copyright.

a. You may seek permission from the original copyright holder of Figure(s) 1A, 1C, 2B, 2D, 3B, 4A,B, C, D, E. 7A, 7B and S4 to publish the content specifically under the CC BY 4.0 license. 

Reviewers' comments:

Reviewer's Responses to Questions

**Comments to the Author**

1. Is the manuscript technically sound, and do the data support the conclusions?

Reviewer #1: Yes

Reviewer #2: Yes

2. Has the statistical analysis been performed appropriately and rigorously? 

Reviewer #1: I Don't Know

Reviewer #2: Yes

3. Have the authors made all data underlying the findings in their manuscript fully available?

Reviewer #1: Yes

Reviewer #2: Yes

4. Is the manuscript presented in an intelligible fashion and written in standard English?

Reviewer #1: Yes

Reviewer #2: Yes

5. Review Comments to the Author

Reviewer #1: Figures:

Comment 1.

In Figure 4D and E, proOLV::GUS activity was induced by -Fe treatment in root differentiation zone and hair. Did the authors test the transcript level of OLV by qPCR at same developmental stage to show that induction is relevant? Seems the expression of OLV is not increased by -Fe in WT (see Fig S5 C).

Comment 2.

In Figure 4F and G shows, proPYE::GUS mainly detected in root tissue only. Why proPYE::GUS activity did not induce in leaves (or shoot) although PYE expression was highly induced both root and shoot by -Fe in Fig 5C (in WT). How do you explain?

Comment 3.

Different letters need improving in Fig5 and Fig6 diagrams. For instance, same letter indicated for all samples in Figure 6F. Seems there are some significant differences between treatment. Similar other errors: Fig5C “ab”, Fig5C “abc”, Fig6B “ab”, Fig6C “ab”.

Comment 4.

Y axis needs improving in Fig 5A and Fig 6A. Please double check.

Comment 5.

Authors used two independent OLV overexpression lines (OX7 and OX11) in all experiments. Expression of OLV was not increased in OX11 (see Fig S5 C and D).

Methods:

Comment 6.

What type of pye-1 mutant was used. Describe it in Plant material section.

Comment 7.

Please, complete the description of protocol. What kind of secondary antibody was used.

Typo errors:

Typographical error in “rot” instead of root (Page9)

Typographical error in “bHLHIVc” (Page5)

Typographical error in “wasused” (Page15)

Typographical error in Fig S4 legend “Subcelllar”.

Reviewer #2: Article review

1. EVALUATION OF THE PAPER MANUSCRIPT

Title of the Manuscript:

"The small iron-deficiency-induced protein OLIVIA and its relation to the bHLH

transcription factor POPEYE"

Manuscript number: PONE-D-23-39597

The article PONE-D-23-39597 has comprehensively demonstrated that Fe deficiency-induced bHLH transcription factor (POPEYE) can bind to the small protein OLIVIA (OLV), which is induced by Fe scarcity. The article PONE-D-23-39597 is slightly suited to the PloS One as per its aims and scope. In general, the manuscript is written in good logical style with a suitable introduction, interesting and reasonable Results and Discussion sections. Therefore, the manuscript can be conditionally accepted. However, to improve the quality of the article, some minor points should be noted:

- Could the authors re-analyze RNA-Seq datasets to see how the OLV gene is expressed in different organs/tissues during growth and development? If so, please write several sentences describing the expression level of the OLV gene.

- The authors should include more information about the OLV in the conclusion section. Also, if possible, please simplify and combine the perspectives into a single paragraph.

- All abbreviations should be applied after the first occurrence, such as transcription factor (pages 5 and 11).

- List of references should be exactly followed the PloS One’s style.

- A minor issue is that the reviewer is unable to access the link to jmicrovision. If possible, offer a new link.

6. PLOS authors have the option to publish the peer review history of their article (what does this mean?). If published, this will include your full peer review and any attached files.

Reviewer #1: No

Reviewer #2: **Yes: **Ha Duc Chu

---

## [Author Response · Author response to Decision Letter 0]

7 Feb 2024

Rebuttal Letter

Dear Editor and reviewers,

We thank you for appreciating our manuscript and very thoroughly studying it to make valuable comments. Please find below our answers to your comments and we hope that we could address them appropriately.

Thank you for your time and looking forward to your response,

Sincerely

Petra Bauer

Our response: To the best of our knowledge, our manuscript now meets the PLoS One style requirements.

Our response: It is not requested/needed that the type of data are deposited in a repository. The blot images were submitted.

 [This work was funded by Deutsche Forschungsgemeinschaft (DFG, German Research Foundation) under GRK F020512056 (NEXTplant) and Germany´s Excellence Strategy – EXC-2048/1 – project ID 390686111. Funding for instrumentation: Zeiss LSM780 + 4-channel FLIM extension (Picoquant): DFG- INST 208/551-1 FUGG.]. 

Our response: This work was funded by Deutsche Forschungsgemeinschaft (DFG, German Research Foundation) under GRK F020512056 (NEXTplant) and Germany´s Excellence Strategy – EXC-2048/1 – project ID 390686111. Funding for instrumentation: Zeiss LSM780 + 4-channel FLIM extension (Picoquant): DFG- INST 208/551-1 FUGG. We state that ""The funders had no role in study design, data collection and analysis, decision to publish, or preparation of the manuscript."" 

The information about funding must appear in the manuscript. Please add it where appropriate.

5. In the online submission form, you indicated that [Data will be published. Seeds and original data can be made available upon request.]. 

Our response: To our knowledge, all data is available in the manuscript, in processed form. We have not added the original unprocessed data.

Our response: We have provided the original blots as supporting information file.

7. We note that Figure(s) 1A, 1C, 2B, 2D, 3B, 4A,B, C, D, E. 7A, 7B and S4 in your submission contain copyrighted images. All PLOS content is published under the Creative Commons Attribution License (CC BY 4.0), which means that the manuscript, images, and Supporting Information files will be freely available online, and any third party is permitted to access, download, copy, distribute, and use these materials in any way, even commercially, with proper attribution. For more information, see our copyright guidelines: http://journals.plos.org/plosone/s/licenses-and-copyright.

a. You may seek permission from the original copyright holder of Figure(s) 1A, 1C, 2B, 2D, 3B, 4A,B, C, D, E. 7A, 7B and S4 to publish the content specifically under the CC BY 4.0 license. 

Our response: In our view our figures of the preprint on biorxiv are available under CC BY 4.0 licence. Please let us know if that is not sufficient. bioRxiv preprint doi: https://doi.org/10.1101/2023.12.05.570225; this version posted December 6, 2023. The copyright holder for this preprint (which was not certified by peer review) is the author/funder, who has granted bioRxiv a license to display the preprint in perpetuity. It is made available under aCC-BY 4.0 International license.

Our response: We do not think, there are copy-righted figures. Please instruct.

Our response: The copyright holder for this preprint is the author/funder, who has granted bioRxiv a license to display the preprint in perpetuity. It is made available under a CC-BY 4.0 International license.

Please let us know if there is anything unclear with it.

We added the legends to supplemental data files at the end of the manuscript.

Comments to the Author

Reviewer #1: 

Figures:

Comment 1.

In Figure 4D and E, proOLV::GUS activity was induced by -Fe treatment in root differentiation zone and hair. Did the authors test the transcript level of OLV by qPCR at same developmental stage to show that induction is relevant? Seems the expression of OLV is not increased by -Fe in WT (see Fig S5 C).

Our response: OLV is co-expressed with iron deficiency-induced genes. In Fig. S5C, the focus is on strong expression in the OLV OX lines, and the representation does not depict differences in OLV expression in the wild type as it is represented along with the overexpressors. In Fig. S5F, there is an induction of OLV expression upon -Fe versus +Fe which is significant in pye. In Fig. S5G, there is a significant induction of OLV in WT and pye.

Comment 2.

In Figure 4F and G shows, proPYE::GUS mainly detected in root tissue only. Why proPYE::GUS activity did not induce in leaves (or shoot) although PYE expression was highly induced both root and shoot by -Fe in Fig 5C (in WT). How do you explain?

Our response: OLV is expressed primarily in roots, and we have been interested in the comparison with PYE expression in roots. The utilized GUS assay procedure works well for root staining. For cotyledon staining, infiltration, cuttings or longer incubation times may have to be used to make GUS substrate penetrate. Please also note that GUS staining is not quantitative. We added: The lines were grown for six days under sufficient (+Fe) or deficient (–Fe) Fe supply and analyzed for promoter-GUS activity in roots (Figure 4D, E, F, G).

Comment 3.

Different letters need improving in Fig5 and Fig6 diagrams. For instance, same letter indicated for all samples in Figure 6F. Seems there are some significant differences between treatment. Similar other errors: Fig5C “ab”, Fig5C “abc”, Fig6B “ab”, Fig6C “ab”.

Our response: We have checked. The representation is correct. 

Comment 4.

Y axis needs improving in Fig 5A and Fig 6A. Please double check.

Our response: Thank you, we corrected it.

Comment 5.

Authors used two independent OLV overexpression lines (OX7 and OX11) in all experiments. Expression of OLV was not increased in OX11 (see Fig S5 C and D).

Our response: Thank you. There is moderate overexpression. The difference to wild type has not been found significant. Fig. S5B shows that there is HA3-OLV protein. We now write: At first, we assessed a possible impact of OLV overexpression (meaning expression behind the double cauliflower mosaic virus 35S promoter).

Methods:

Comment 6.

What type of pye-1 mutant was used. Describe it in Plant material section.

Our response: Thank you, we added it.

Comment 7.

Please, complete the description of protocol. What kind of secondary antibody was used.

Our response: anti-HA high-affinity monoclonal rat antibody - conjugated with horseradish peroxidase (clone 3F10; Roche), a secondary antibody is not needed.

Typo errors:

Typographical error in “rot” instead of root (Page9)

Typographical error in “bHLHIVc” (Page5)

Typographical error in “wasused” (Page15)

Typographical error in Fig S4 legend “Subcelllar”.

Our response: Thank you, was done.

Reviewer #2: Article review

1. EVALUATION OF THE PAPER MANUSCRIPT

Title of the Manuscript:

"The small iron-deficiency-induced protein OLIVIA and its relation to the bHLH

transcription factor POPEYE"

Manuscript number: PONE-D-23-39597

The article PONE-D-23-39597 has comprehensively demonstrated that Fe deficiency-induced bHLH transcription factor (POPEYE) can bind to the small protein OLIVIA (OLV), which is induced by Fe scarcity. The article PONE-D-23-39597 is slightly suited to the PloS One as per its aims and scope. In general, the manuscript is written in good logical style with a suitable introduction, interesting and reasonable Results and Discussion sections. Therefore, the manuscript can be conditionally accepted. However, to improve the quality of the article, some minor points should be noted:

- Could the authors re-analyze RNA-Seq datasets to see how the OLV gene is expressed in different organs/tissues during growth and development? If so, please write several sentences describing the expression level of the OLV gene.

Our response: Thank you, we added: According to available transcriptome data under control conditions, OLV is expressed during germination and in the root epidermis (Klepikova et al., 2016; Ryu et al., 2019).

- The authors should include more information about the OLV in the conclusion section. Also, if possible, please simplify and combine the perspectives into a single paragraph.

Our response: Thank you, we shortened the conclusions/perspectives paragraph: OLV is a small protein interactor of PYE that may interact via a conserved previously non-characterized TGIYY motif in the Fe deficiency response. OLV had a mild effect on plant growth and Fe deficiency responses involving NAS4. Since NAS4 is specifically targeted by other TFs of the Fe deficiency response, OLV may control a particular sub-function of PYE in controlling NAS4. OLV is primarily expressed during germination and in in roots where it is induced by Fe deficiency. Future studies can address the functional and structural relevance of the TGIYY motif for formation of the protein complex with PYE. Protein interaction can be further studied in plant roots, e.g. by co-expressing fusion proteins of OLV and PYE expressed from their respective promoters. 

- All abbreviations should be applied after the first occurrence, such as transcription factor (pages 5 and 11).

Our response: Thank you, was done.

- List of references should be exactly followed the PloS One’s style.

Our response: Is now the correct style.

- A minor issue is that the reviewer is unable to access the link to jmicrovision. If possible, offer a new link.

Our response: Thank you, we corrected it.

---

## [Decision Letter · Decision Letter 1]

11 Mar 2024

PONE-D-23-39597R1The small iron-deficiency-induced protein OLIVIA and its relation to the bHLH transcription factor POPEYEPLOS ONE

Dear Dr. Bauer,

Thank you for submitting your manuscript to PLOS ONE. After careful consideration, we feel that it has merit but does not fully meet PLOS ONE’s publication criteria as it currently stands. Therefore, we invite you to submit a revised version of the manuscript that addresses the points raised during the review process.

We look forward to receiving your revised manuscript.

Kind regards,

Nguyen Hoai Nguyen

Academic Editor

PLOS ONE

Journal Requirements:

**Additional Editor Comments:**

I also agree with the Reviewer 1 regarding the usage of biomass in this case. Please amend your writing in the manuscript according to the obtained data (e.g., plant size...). 

Reviewers' comments:

Reviewer's Responses to Questions

**Comments to the Author**

1. If the authors have adequately addressed your comments raised in a previous round of review and you feel that this manuscript is now acceptable for publication, you may indicate that here to bypass the “Comments to the Author” section, enter your conflict of interest statement in the “Confidential to Editor” section, and submit your "Accept" recommendation.

Reviewer #1: (No Response)

Reviewer #2: All comments have been addressed

2. Is the manuscript technically sound, and do the data support the conclusions?

Reviewer #1: Yes

Reviewer #2: Yes

3. Has the statistical analysis been performed appropriately and rigorously? 

Reviewer #1: Yes

Reviewer #2: Yes

4. Have the authors made all data underlying the findings in their manuscript fully available?

Reviewer #1: Yes

Reviewer #2: Yes

5. Is the manuscript presented in an intelligible fashion and written in standard English?

Reviewer #1: Yes

Reviewer #2: Yes

6. Review Comments to the Author

Reviewer #1: Comment1:

In page 5, you mentioned “OLV also affects biomass”. Indeed, olv mutant showed smaller phenotype compared to WT (supported by rosette size data), but you do not have fresh weight value. Please, think on that, or just remove “biomass”.

Typo errors:

Typographical error in “BHLH” (Page5). Usually, we use bHLH.

Typographical error in “Sesanum indicum” (Page 7).

Typographical errors in FigS2. Please, double check the name of genus: “Actinida chinensis “, “Sesamum idicum”, “Viits vinifera”.

Typographical errors in FigS3. Please, double check “Sesamun indicum “, “Gingko biloba”.

Typographical error in FigS4 “Subcelllar”.

Reviewer #2: All comments have been addressed. The reviewer believes that this manuscript version meets the requirements for publication in PloS ONE.

7. PLOS authors have the option to publish the peer review history of their article (what does this mean?). If published, this will include your full peer review and any attached files.

Reviewer #1: **Yes: **Shapulatov Umidjon

Reviewer #2: No

---

## [Author Response · Author response to Decision Letter 1]

12 Mar 2024

Cover Letter

Dear Editor and reviewers,

We thank you for appreciating our revised manuscript and for very thoroughly studying the revision to make valuable comments. Please find below our answers to your comments. All corrections were made and we hope that we could address them appropriately. 

Thank you for your time and looking forward to your response,

Hopefully our manuscript is now ready for acceptance,

Sincerely

Petra Bauer

Reviewrs' comments:

Reviewer's Responses to Questions

Comments to the Author

1. If the authors have adequately addressed your comments raised in a previous round of review and you feel that this manuscript is now acceptable for publication, you may indicate that here to bypass the “Comments to the Author” section, enter your conflict of interest statement in the “Confidential to Editor” section, and submit your "Accept" recommendation.

Reviewer #1: (No Response)

Reviewer #2: All comments have been addressed

2. Is the manuscript technically sound, and do the data support the conclusions?

Reviewer #1: Yes

Reviewer #2: Yes

3. Has the statistical analysis been performed appropriately and rigorously? 

Reviewer #1: Yes

Reviewer #2: Yes

4. Have the authors made all data underlying the findings in their manuscript fully available?

Reviewer #1: Yes

Reviewer #2: Yes

5. Is the manuscript presented in an intelligible fashion and written in standard English?

Reviewer #1: Yes

Reviewer #2: Yes

6. Review Comments to the Author

Reviewer #1: Comment1:

In page 5, you mentioned “OLV also affects biomass”. Indeed, olv mutant showed smaller phenotype compared to WT (supported by rosette size data), but you do not have fresh weight value. Please, think on that, or just remove “biomass”.

We exchanged “biomass” against “rosette size”.

Typo errors:

Typographical error in “BHLH” (Page5). Usually, we use bHLH.

bHLH designates a protein; BHLH designates DNA/RNA/gene. We checked for correctness throughout the text.

Typographical error in “Sesanum indicum” (Page 7).

corrected 

Typographical errors in FigS2. Please, double check the name of genus: “Actinida chinensis “, “Sesamum idicum”, “Viits vinifera”.

corrected 

Typographical errors in FigS3. Please, double check “Sesamun indicum “, “Gingko biloba”.

corrected 

Typographical error in FigS4 “Subcelllar”.

corrected 

Our response: The authors thank reviewer 1 for suggestions. We corrected all of the spelling mistakes. 

Reviewer #2: All comments have been addressed. The reviewer believes that this manuscript version meets the requirements for publication in PloS ONE.

Our response: The authors thank reviewer 2 for assessment.

7. PLOS authors have the option to publish the peer review history of their article (what does this mean?). If published, this will include your full peer review and any attached files.

Do you want your identity to be public for this peer review? For information about this choice, including consent withdrawal, please see our Privacy Policy.

Reviewer #1: Yes: Shapulatov Umidjon

Reviewer #2: No

---

## [Editor Report · Decision Letter 2]

22 Mar 2024

The small iron-deficiency-induced protein OLIVIA and its relation to the bHLH transcription factor POPEYE

PONE-D-23-39597R2

Dear Dr. Bauer,

We’re pleased to inform you that your manuscript has been judged scientifically suitable for publication and will be formally accepted for publication once it meets all outstanding technical requirements.

Kind regards,

Nguyen Hoai Nguyen

Academic Editor

PLOS ONE
---

## [Editor Report · Acceptance letter]

3 Apr 2024

PONE-D-23-39597R2 

PLOS ONE

Dear Dr. Bauer, 

I'm pleased to inform you that your manuscript has been deemed suitable for publication in PLOS ONE. Congratulations! Your manuscript is now being handed over to our production team.

Kind regards, 

on behalf of

Dr. Nguyen Hoai Nguyen 

Academic Editor

PLOS ONE